# Inhibition of the Notch signal transducer CSL by Pkc53E-mediated phosphorylation to fend off parasitic immune challenge in *Drosophila*

Sebastian Deichsel[1,2], Lisa Frankenreiter[1], Johannes Fechner[1,3], Bernd M Gahr[1,4], Mirjam Zimmermann[1], Helena Mastel[1], Irina Preis[1], Anette Preiss[1], Anja C Nagel[1]*

[1]Department of Molecular Genetics, Institute of Biology, University of Hohenheim, Stuttgart, Germany; [2]Department of Medical Genetics and Applied Genomics, University of Tübingen, Tübingen, Germany; [3]Institute of Biomedical Genetics (IBMG), University of Stuttgart, Stuttgart, Germany; [4]Department of Internal Medicine II, Molecular Cardiology, University of Ulm, Ulm, Germany

## eLife Assessment

This **valuable** study focuses on the regulation of Notch signaling during the immune response in Drosophila. The authors provide **solid** evidence in support of roles for Su(H) and Pkc53E-induced phosphorylation in *Drosophila* immunity. The work will be of interest to colleagues in immunity and receptor signaling.

**\*For correspondence:**
anja.nagel@uni-hohenheim.de

**Competing interest:** The authors declare that no competing interests exist.

**Abstract** Notch signalling activity regulates hematopoiesis in *Drosophila* and vertebrates alike. Parasitoid wasp infestation of *Drosophila* larvae, however, requires a timely downregulation of Notch activity to allow the formation of encapsulation-active blood cells. Here, we show that the *Drosophila* CSL transcription factor Suppressor of Hairless [Su(H)] is phosphorylated at Serine 269 in response to parasitoid wasp infestation. As this phosphorylation interferes with the DNA binding of Su(H), it reversibly precludes its activity. Accordingly, phospho-deficient *Su(H)*^S269A mutants are immune-compromised. A screen for kinases involved in Su(H) phosphorylation identified Pkc53E, required for normal hematopoiesis as well as for parasitoid immune response. Genetic and molecular interactions support the specificity of the Su(H)-Pkc53E relationship. Moreover, phorbol ester treatment inhibits Su(H) activity in vivo and in human cell culture. We conclude that Pkc53E targets Su(H) during parasitic wasp infestation, thereby remodelling the blood cell population required for wasp egg encapsulation.

## Introduction

*Drosophila melanogaster* harbours a sophisticated cellular immune system to fight invaders. The larval hematopoietic system comprises a circulating and sessile compartment of embryonic origin and the developing larval lymph gland, constituting a hematopoietic organ. The circulating and sessile compartment consists primarily of macrophage-like plasmatocytes, plus a small number of crystal cells involved in wound healing and melanisation responses to neutralise pathogens. Both cell types differentiate from hemocyte precursors within the lymph gland as well, and they are released during pupal stages to serve the adult with immune cells. Finally, lamellocytes represent the third blood cell type

in *Drosophila*. While merely absent in healthy animals, their formation is induced within hours of wasp infestation or wounding (reviewed in *Banerjee et al., 2019*; *Letourneau et al., 2016*; *Hultmark and Andó, 2022*). In fact, parasitism by endo-parasitoid wasps represents one of the most severe naturally occurring immune challenges to *Drosophila*, invariably causing death if not defended off properly. Parasitoid wasps deposit their egg into the live *Drosophila* larva, where it develops into the next wasp generation egressing from the pupa instead of a fly. Hence, wasp infestation is a life-threatening challenge to the *Drosophila* host, demanding an immediate immune response to overwhelm the parasite. This involves a massive increase in circulating hemocytes, and most energy resources are devoted to fight the invader. Here, lamellocytes play a critical role: they encapsulate the wasp egg and, by expressing prophenoloxidase, induce a melanisation reaction retarding further wasp development (*Dudzic et al., 2015*; reviewed in: *Banerjee et al., 2019*; *Letourneau et al., 2016*; *Hultmark and Andó, 2022*).

Wasp infestation substantially remodels the composition of the *Drosophila* hemocyte population (*Cattenoz et al., 2020*; *Cho et al., 2020*; *Tattikota et al., 2020*; reviewed in *Csordás et al., 2021*). There is a vast increase in plasmatocytes and intermediate precursors in both hematopoietic compartments, from which lamellocyte differentiate. This process requires the combined regulatory input of several pathways, including JAK/STAT, JNK, Toll, Notch, EGFR, and INR pathways, which in fact regulate blood cell homeostasis in general (reviewed in *Banerjee et al., 2019*; *Letourneau et al., 2016*; *Csordás et al., 2021*). Simultaneous to the massive expansion of lamellocytes, crystal cells are significantly reduced (*Crozatier et al., 2004*; *Krzemien et al., 2010*; *Ferguson and Martinez-Agosto, 2014*; *Cattenoz et al., 2020*; *Cho et al., 2020*; *Tattikota et al., 2020*; reviewed in *Csordás et al., 2021*). Crystal cells are generated both in the larval lymph gland and by transdifferentiation from plasmatocytes in the sessile compartment. Their formation, differentiation, and survival strictly depend on Notch signalling activity (reviewed in *Banerjee et al., 2019*; *Csordás et al., 2021*; *Hultmark and Andó, 2022*). Lamellocyte and crystal cell lineages are mutually exclusive. Accordingly, while promoting crystal cell fate, Notch activity inhibits differentiation of lamellocytes within the lymph gland, however, is reduced upon wasp infestation (*Small et al., 2014*). Lamellocyte induction involves the formation and sensing of reactive oxygen species triggered by wasp egg injection (*Nappi et al., 1995*; *Sinenko et al., 2011*; *Small et al., 2014*; *Louradour et al., 2017*; reviewed in *Banerjee et al., 2019*; *Csordás et al., 2021*). The molecular mechanism underlying the simultaneous crystal cell fate inhibition, however, is less well understood. Obviously, an effective and timely downregulation of Notch activity is required to fend off parasitic wasps.

The Notch pathway is highly conserved between invertebrates and vertebrates, where it regulates numerous cell fate decisions. Notch signals are transduced by CSL-type proteins (abbreviation of human C̲BF1/RBPJ, *Drosophila* S̲uppressor of Hairless [Su(H)], and worm L̲ag1), that bind the DNA to direct transcriptional activity of Notch target genes by help of recruited cofactors (reviewed in *Giaimo et al., 2021*; *Kopan and Ilagan, 2009*; *Siebel and Lendahl, 2017*). In the absence of Notch signals, however, CSL together with co-repressors silences Notch target genes, thereby acting as a molecular switch (*Borggrefe and Oswald, 2009*). Hence, CSL is central to Notch pathway activity as no signal transduction can occur in its absence or in the instance of a lack of DNA binding. Earlier, we observed Su(H) phosphorylation at Serine 269 in cultured *Drosophila* Schneider S2 cells (*Nagel et al., 2017*). Of note, Schneider S2 cells have hemocyte characteristics (*Cherbas et al., 2011*; *Schneider, 1972*; *Terriente-Felix et al., 2013*). As a consequence of the negative charge conferred by the phosphorylation at Serine 269, Su(H) loses its affinity to the DNA, and without its DNA-binding ability, also its function as a transcriptional regulator (*Nagel et al., 2017*). Accordingly, a phospho-mimetic *Su(H)*$^{S269D}$ mutant behaved like a *Su(H)* loss-of-function mutant in all respects. In contrast, the phospho-deficient *Su(H)*$^{S269A}$ variant, appeared wild-type at the first glance, indicating some tissue specificity of this regulatory mechanism. In fact, the *Su(H)*$^{S269A}$ allele displayed a gain of Notch activity particularly during embryonic and larval hematopoiesis with increased numbers of crystal cells in both hematopoietic compartments (*Frankenreiter et al., 2021*). Moreover, we found that the general Notch antagonist Hairless is not involved in constraining crystal cell numbers, suggesting that in the context of blood cell homeostasis Notch activity is regulated by the phosphorylation of Su(H) (*Maier, 2006*; *Frankenreiter et al., 2021*). As the same set of genes affect blood cell maintenance and differentiation in homeostasis and upon immune challenge (*Cho et al., 2020*; *Tattikota et al., 2020*), obviously phospho-mediated downregulation of Notch activity might also occur in response to parasitoid wasp

infestation, allowing the formation of encapsulation-active lamellocytes at the expense of crystal cells. Briefly, phosphorylation of Su(H) is an elegant mechanism to transiently curb Notch activity in the context of immune responses.

In this work, we followed the hypothesis that after wasp infestation, a specific kinase might be activated to phosphorylate Su(H) thereby allowing an adequate immune response. In this case, the phospho-deficient $Su(H)^{S269A}$ allele should be immune-compromised, as it cannot respond to phosphorylation, i.e., remaining active even upon parasitism. Indeed, $Su(H)^{S269A}$ displayed an increased sensitivity towards parasitoid wasp infestation accompanied by an increase of crystal cells at the expense of lamellocytes. Accordingly, phosphorylation of Su(H) protein was detected in infested wild-type larvae but not in the $Su(H)^{S269A}$ mutant. In a screen for kinases regulating this process, Pkc53E, the homologue of human PKCα, was identified as an important player. In agreement with a role in blood cell homeostasis, a $Pkc53E^{\Delta 28}$ null mutant displayed increased crystal cell numbers. Moreover, genetic and molecular interactions between Su(H) and Pkc53E support the specificity of their relationship. Finally, $Pkc53E^{\Delta 28}$ was impaired in its immune response to wasp infestation as well. Together, these data show that Su(H) is a target of Pkc53E during parasitic wasp infestation, inducing phosphorylation and subsequent downregulation of Su(H) activity to allow the mass production of lamellocytes required for wasp defense.

## Results

### Impaired immune response to parasitoid wasps in the phospho-deficient Su(H)^S269A^ mutant

Wasp infestation alters the course of hematopoiesis, as lamellocyte differentiation is massively increased at the expense of crystal cells. This process requires a timely attenuation of Notch activity that may be implemented by the phosphorylation of Su(H) at Serine 269 (S269). To test this model, we exposed control larvae and larvae of the phospho-deficient $Su(H)^{S269A}$ variant to the parasitoid wasp *Leptopilina boulardi* (*L. boulardi*) and analysed the consequences on blood cell homeostasis around 44 hr post-infection, i.e., before lymph gland histolysis. To exclude any influence of the engineered genomic background, we used $Su(H)^{gwt}$ for comparison, carrying a genomic wild-type construct in place of the mutant (*Praxenthaler et al., 2017*; *Frankenreiter et al., 2021*).

Earlier we noted an excess of crystal cells in the phospho-deficient $Su(H)^{S269A}$ allele, which we interpret as a gain of Notch activity in consequence of the inability to be downregulated by Su(H) phosphorylation (*Frankenreiter et al., 2021*; *Figure 1A and B*). In agreement with this hypothesis, RNAi-mediated downregulation of *Notch* in hemocytes (*hml::N*-RNAi) resulted in a near-complete loss of crystal cells. This *Notch* loss-of-function phenotype was epistatic to the $Su(H)^{S269A}$ phenotype, i.e., the excess of crystal cells characterising $Su(H)^{S269A}$ was no longer observed in the combination with *N*-RNAi, demonstrating that Notch acted upstream of Su(H) as expected (*Figure 1—figure supplement 1*). The slightly elevated numbers in the $Su(H)^{S269A}$ background compared to the control, however, may be due to the enlarged anlagen in the embryo unaffected by *hml::N*-RNAi (*Frankenreiter et al., 2021*). Total hemocyte numbers were slightly, albeit not significantly increased in $Su(H)^{S269A}$ compared to the $Su(H)^{gwt}$ control, and correspondingly, hemocyte numbers were somewhat lowered in the $Su(H)^{S269D}$ allele (*Figure 1—figure supplement 1*). This is in line with earlier observations of unchanged plasmatocyte counts in *N* or *Su(H)* mutants relative to control (*Duvic et al., 2002*). In response to wasp infestations, however, crystal cell numbers should drop to allow the formation of lamellocytes (*Small et al., 2014*; *Csordás et al., 2021*). Indeed in the $Su(H)^{gwt}$ control, both the sessile crystal cells and those within the larval lymph glands were significantly lessened in response to wasp infestation (*Figure 1A and B*). In contrast, the higher crystal cell numbers in the $Su(H)^{S269A}$ mutant larvae dropped to control level, demonstrating the impairment of the mutant to detect this immune challenge or to respond to it (*Figure 1A and B*). Total hemocyte numbers, however, were similar between the genotypes independent of wasp infestation (*Figure 1—figure supplement 1*).

Increased abundance of lamellocytes upon wasp infestation was monitored in vivo in larval hemolymph and lymph glands, using either the L1-*atilla*-GFP reporter (*Honti et al., 2009*) or *PPO3*-Gal4::UAS-GFP (*Dudzic et al., 2015*), respectively. In the harvested larval hemolymph of the infested $Su(H)^{gwt}$ control, the hemocytes contained about 15% *PPO3-* and 23% *atilla*-labelled lamellocytes (*Figure 1C and D*). These numbers were significantly lower in the wasp infested $Su(H)^{S269A}$ larvae

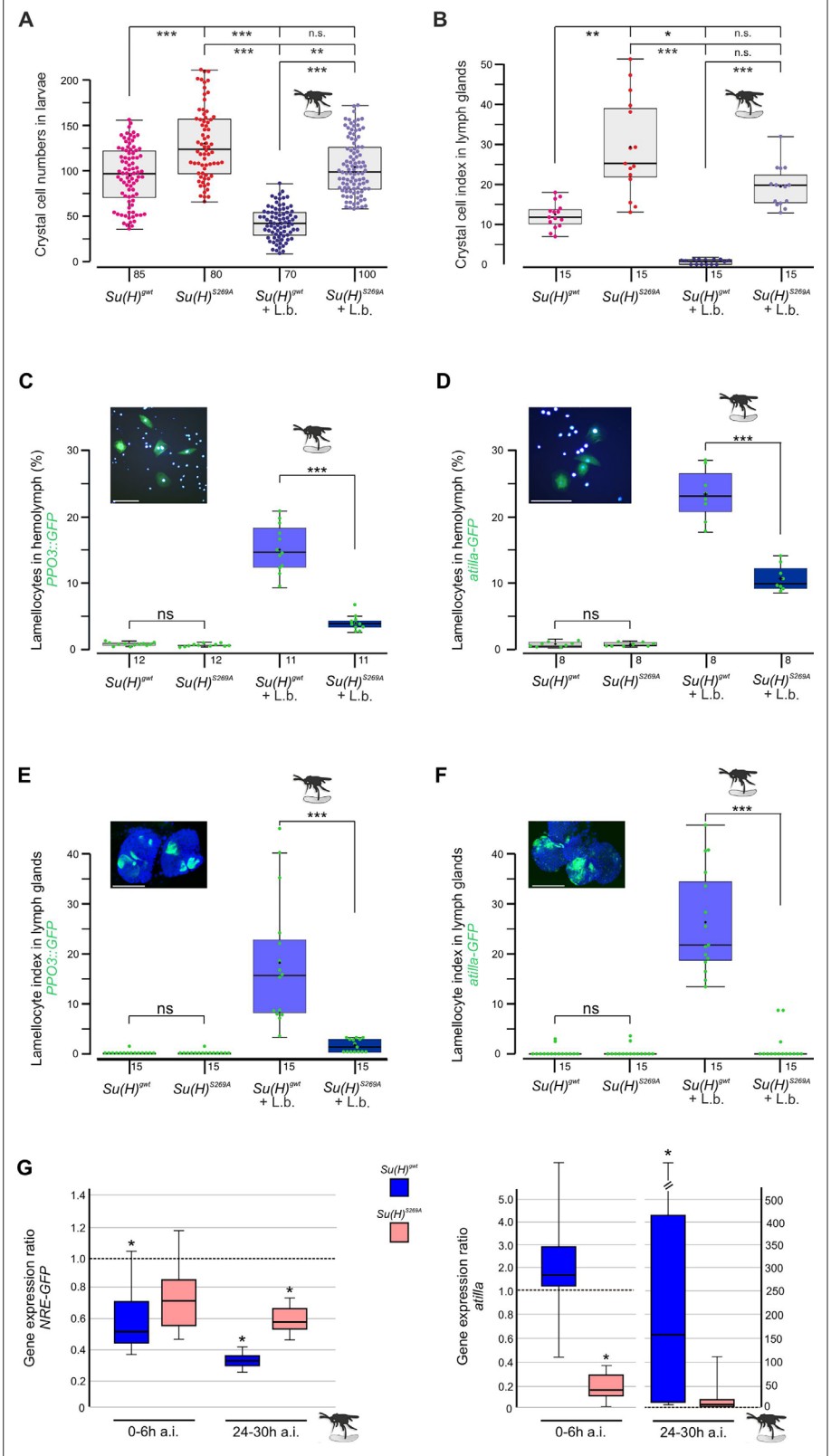

**Figure 1.** Su(H)^S269A mutants are compromised in their response to parasitoid wasp infestation. (**A**) Quantification of melanised crystal cells from the last two segments of Su(H)^gwt and Su(H)^S269A larvae with and without wasp infestation as indicated. In the control, wasp parasitism causes crystal cell numbers to drop to a level of about 50%, whereas in Su(H)^S269A mutants the number settles at the uninfested Su(H)^gwt level. Each dot represents one

*Figure 1 continued on next page*

*Figure 1 continued*

analysed larva (n=70–100 as indicated). (**B**) Crystal cell index in larval lymph glands is given as ratio of Hnt-positive crystal cells per 1° lobe relative to the size of the lobe. Each point represents one analysed lobus (n=15). (**A, B**) Statistical analyses with Kruskal-Wallis test, followed by Dunn's test with ***p<0.001, **p<0.01, *p<0.05, ns (not significant p≥0.05). (**C–F**) Quantification of larval lamellocytes in the circulating hemolymph (**C, D**) or in lymph glands (**E, F**) before and after wasp infestation in *Su(H)$^{gwt}$* versus *Su(H)$^{S269A}$*. Lamellocytes were marked with either *PPO3-Gal4::UAS*-GFP (**C, E**) or *atilla*-GFP (**D, F**) as indicated. (**C, D**) The fraction of GFP-labelled lamellocytes of the total number of DAPI-labelled blood cells isolated from hemolymph is given; each dot represents 10 pooled larvae (n=8–10 as shown). Representative image of labelled control hemolymph is shown above (DAPI-labelled nuclei in light blue, GFP in green). Scale bars, 50 µm. (**E, F**) Lamellocyte index is given as number of GFP-labelled lamellocytes per area in the 1° lobe of the lymph gland. Each dot represents the lamellocyte index of one lobus (n=15). Representative *Su(H)$^{gwt}$* lymph glands after wasp infestation are shown, co-stained for nuclear Pzg (in blue). Scale bars, 100 µm. Statistical analyses with unpaired Student's t-test; only significant differences are indicated (***p<0.001). (**A–F**) Representative images for each genotype and condition are shown in *Figure 1—figure supplement 2*. (**G**) qRT-PCR analyses measuring expression of NRE-GFP (left panel) and *atilla* (right panel). Transcript levels were quantified from hemolymph isolated from infested larvae at 0–6 hr or 24–30 hr post-infestation as indicated, relative to the untreated *Su(H)$^{gwt}$* control. *Tbp* and *cyp33* served as reference genes. Shown data were gained from four biological and two technical replicates each. Left panel: Immediately after wasp infection, NRE-GFP expression dropped significantly in the *Su(H)$^{gwt}$* control, and even further to about 30% 24–30 hr post-infection, whereas it remained at 60–70% in the infested *Su(H)$^{S269A}$* mutants. Right panel: *atilla* transcripts remained stable at first in the *Su(H)$^{gwt}$* control, to rise dramatically 24–30 hr post-infection, in contrast to *Su(H)$^{S269A}$*. Mini-max depicts 95% confidence, mean corresponds to expression ratio. Exact p-values are given in the raw data table. Significance was tested using PFRR from REST (*p<0.05).

The online version of this article includes the following source data and figure supplement(s) for figure 1:

**Source data 1.** Raw data and statistical analysis.

**Figure supplement 1.** Notch acts upstream of Su(H); minor changes in hemocyte numbers in Su(H)$^{S269}$ phospho-mutants.

**Figure supplement 1—source data 1.** Raw data and statistical analysis.

**Figure supplement 2.** Representative images for the various settings.

---

(*Figure 1C and D*). Consistently, both lamellocyte reporters were robustly induced in the lymph glands of the infested control, but rarely in glands of the *Su(H)$^{S269A}$* mutant (*Figure 1E and F* and *Figure 1—figure supplement 2*). Obviously, the *Su(H)$^{S269A}$* mutant barely responds to the immune challenge raised by the parasitic wasp infestation.

In order to monitor the altered immune responses at the molecular level, we quantified Notch target gene expression in the hemolymph upon wasp infestation over time. We observed a decline in the expression of the Notch reporter NRE-GFP immediately after wasp infection in the control *Su(H)$^{gwt}$* to about half the value of the uninfected larvae, dropping even further to about 30% 24–30 hr post-infection (*Figure 1G*). *Su(H)$^{S269A}$* mutant larvae, however, retained a much stronger expression level at around 60–70% of the uninfected control even at the late time point. These data reveal the downregulation of Notch activity in response to wasp infestation prior or parallel to lamellocyte formation, in agreement with our model, whereby the infestation-induced phosphorylation of Su(H) impairs transmission of Notch signalling activity. Accordingly, the lamellocyte-specific marker *atilla* bounced up nearly two magnitudes in the wasp infected *Su(H)$^{gwt}$* control at the late time point, but not in *Su(H)$^{S269A}$* compared to the uninfected control (*Figure 1G*; *Cattenoz et al., 2020*).

Overall, these data support the model that S269 in Su(H) is a molecular target for a kinase, phosphorylated upon immune challenge to allow lamellocyte formation at the expense of crystal cells.

## The phospho-deficient Su(H)$^{S269A}$ allele is impaired in combating wasp infestation

Parasitoid wasp infestation constitutes an extreme immune challenge for the *Drosophila* larva: if not combatted by the immune system, a wasp egg, which is deposited in the larval body cavity, will develop into an adult wasp, thereby killing the larval host during the pupal stage. Indeed, depending on the wasp species used, we measured a high mortality rate with less than 5% up to about 14% of surviving flies, whereas nearly all pupae hatched to adults without wasp challenge (*Figure 2A*). According to our working hypothesis, the phospho-deficient *Su(H)$^{S269A}$* variant should not be able to

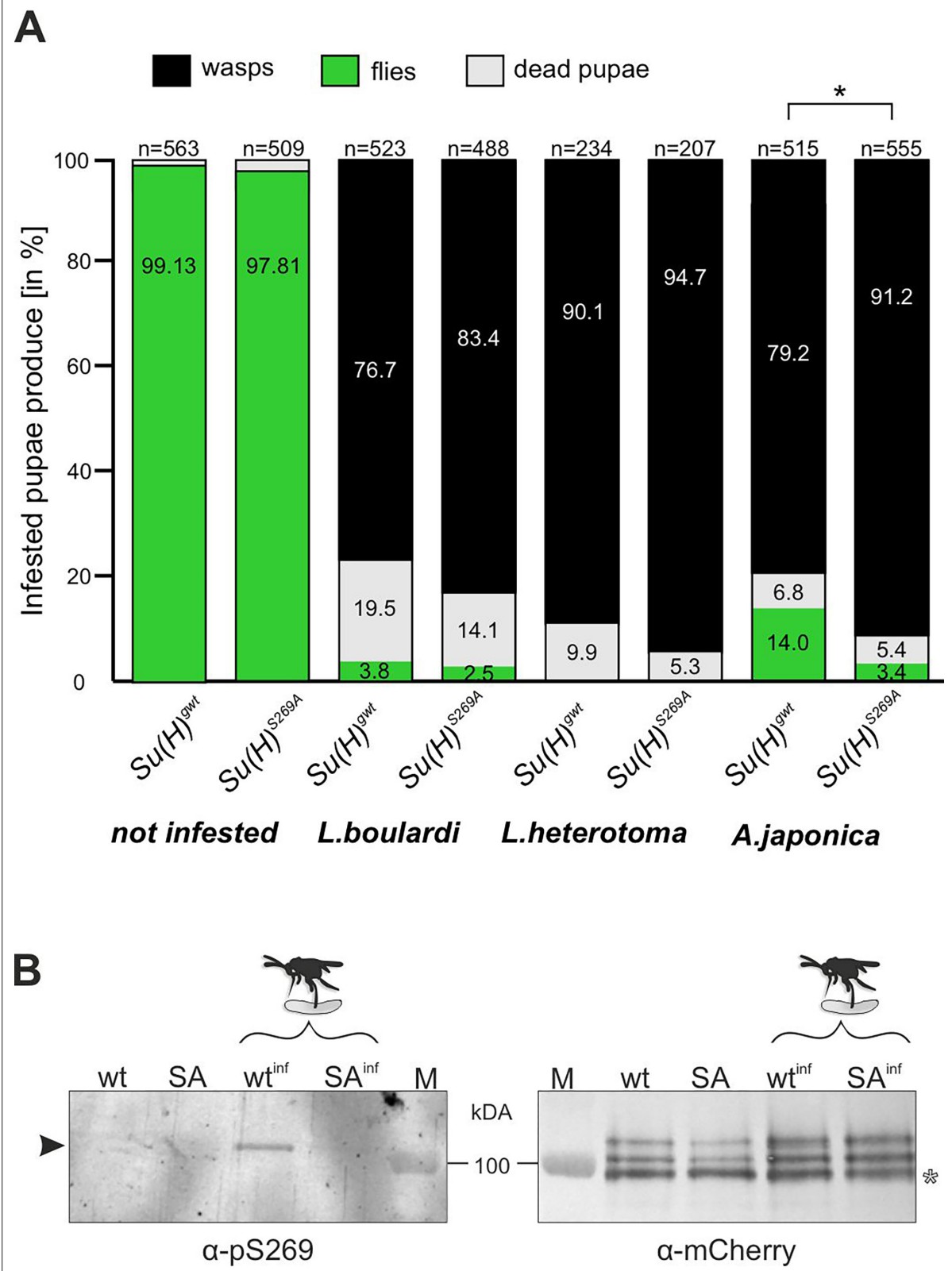

**Figure 2.** Resistance to wasp infestation and phosphorylation at S269. (**A**) Resistance of *Su(H)*<sup>gwt</sup> and *Su(H)*<sup>S269A</sup> to the infestation with parasitic wasp strains *L. boulardi* and *L. heterotoma* (both family Figitidae) and *Asobara japonica* (family Braconidae), as indicated. Numbers of eclosed flies versus wasps as well as of dead pupae are presented in relation to the total of infested pupae. Left two columns are from non-infested controls. At least three independent experiments were performed, n=number of infested pupae. Statistically significant difference determined by Student's t-test is indicated

*Figure 2 continued on next page*

*Figure 2 continued*

with \*p<0.05. (**B**) Su(H) is phosphorylated at Serine 269 upon parasitoid wasp infestation. Protein extracts from *Su(H)*$^{gwt-mCh}$ (wt) and *Su(H)*$^{S269A-mCh}$ (SA) larvae, respectively, infested (wt$^{inf}$, SA$^{inf}$) with *L. boulardi* or uninfested, were isolated by RFP-Trap precipitation and probed in western blots. The anti-pS269 antiserum specifically detects wild-type Su(H) protein only in wasp infested larvae (arrowhead), but not the Su(H)$^{S269A}$ isoform. The blot on the right served as loading control, probed with anti-mCherry antibodies, revealing the typical Su(H) protein pattern in all lanes; the lowest band presumably stems from degradation (open asterisk). M, prestained protein ladder, protein size is given in kDa.

The online version of this article includes the following source data and figure supplement(s) for figure 2:

**Source data 1.** Original, uncropped western blots shown in *Figure 2B*, probed with anti-pS269 and anti-mCherry, respectively.

**Source data 2.** Original, uncropped western blots shown in *Figure 2B*, probed with anti-pS269 and anti-mCherry, respectively - with labelling.

**Source data 3.** Raw data and statistical analysis.

**Figure supplement 1.** The α-pS269 antiserum detects the phospho-mimetic Su(H) variant in vitro.

**Figure supplement 1—source data 1.** Original, uncropped western blot.

**Figure supplement 1—source data 2.** Original, uncropped western blot - labelled.

properly respond to the immune challenge. To test this directly, we measured the survival of *Su(H)*$^{S269A}$ animals upon wasp infestation compared to the wild-type control.

The two closely related wasp species *L. boulardi* and *Leptopilina heterotoma* (*L. heterotoma*) very efficiently parasitised both the control *Su(H)*$^{gwt}$ and the *Su(H)*$^{S269A}$ variant, though the latter appeared slightly more sensitive (*Figure 2A*). The difference in mortality became more apparent with the wasp species *Asobara japonica* (*A. japonica*), allowing 14% of the *Su(H)*$^{gwt}$ control flies to escape parasitism, whereas only 3.4% of the infested *Su(H)*$^{S269A}$ pupae emerged as flies. Thus, the *Su(H)*$^{S269A}$ mutants are less robust in resisting parasitoid wasp infestation consistent with an impaired immune response.

## Serine 269 of Su(H) is phosphorylated upon wasp infestation

Next, we wanted to directly monitor Su(H) phosphorylation at S269 in response to wasp infestation. To this end, polyclonal antibodies directed against a phosphorylated peptide containing the sequence motif NRLRpSQTVSTRYLHVE were generated (α-pS269). Specificity was first tested by western blot analysis using bacterially expressed GST fusion proteins containing the entire beta-trefoil domain (BTD) of Su(H), as well as with phospho-mimetic (S269D) and phospho-mutant (S269A) versions. All three variants were detected by the antisera. The S269D version, however, was strongly preferred, indicating that this antibody does preferably recognise phospho-S269 Su(H) protein (*Figure 2—figure supplement 1*). Encouraged by this result, we used this antiserum on protein extracts derived from larvae infested and not infested by *L. boulardi*. For this experiment we used genome engineered fly strains expressing mCherry-tagged Su(H) proteins, *Su(H)*$^{S269A-mCh}$ and *Su(H)*$^{gwt-mCh}$ for control (*Praxenthaler et al., 2017*). Su(H) proteins were trapped using RFP-nanobody-coupled agarose beads, and the precipitates were then probed in western blots with antibodies directed against mCherry and pS269. Indeed, α-pS269 antibodies recognised Su(H)$^{gwt-mCh}$ protein in wasp infested larvae, indicating respective phosphorylation of Su(H) protein. No such signals were seen in precipitates from the non-infected larvae nor from any of the *Su(H)*$^{S269A-mCh}$ mutant larvae (*Figure 2B*). These data strongly support the notion of parasitism-induced phosphorylation of Su(H) at S269. Apparently, parasitoid wasp infestation starts a cascade of events resulting in the inhibitory S269 phosphorylation of Su(H) protein to allow lamellocyte formation. Hence, the question arose on the kinase/s involved in this process.

## Screening of kinase candidates mediating phosphorylation of Su(H) at Serine 269

To identify Ser/Thr kinases involved in the phosphorylation of Su(H) at S269, we commenced with a combination of in silico and biochemical approaches aiming to generate a list of candidate kinases which can be further analysed by genetic means (*Figure 3*). First, by using GPS 3.0 software that encompasses a substantial database of kinases and their preferred recognition motifs (*Xue et al., 2011*), 36 potential human kinases were predicted to recognise S269 as substrate, represented by 30 kinases in *Drosophila* (*Supplementary file 1*). In addition, the BTD domain of Su(H) was bacterially expressed as a GST fusion protein and subjected to phospho-assays using 245 different human Ser/Thr kinases. Our reasoning for using the entire BTD domain was to ensure a normal folding of

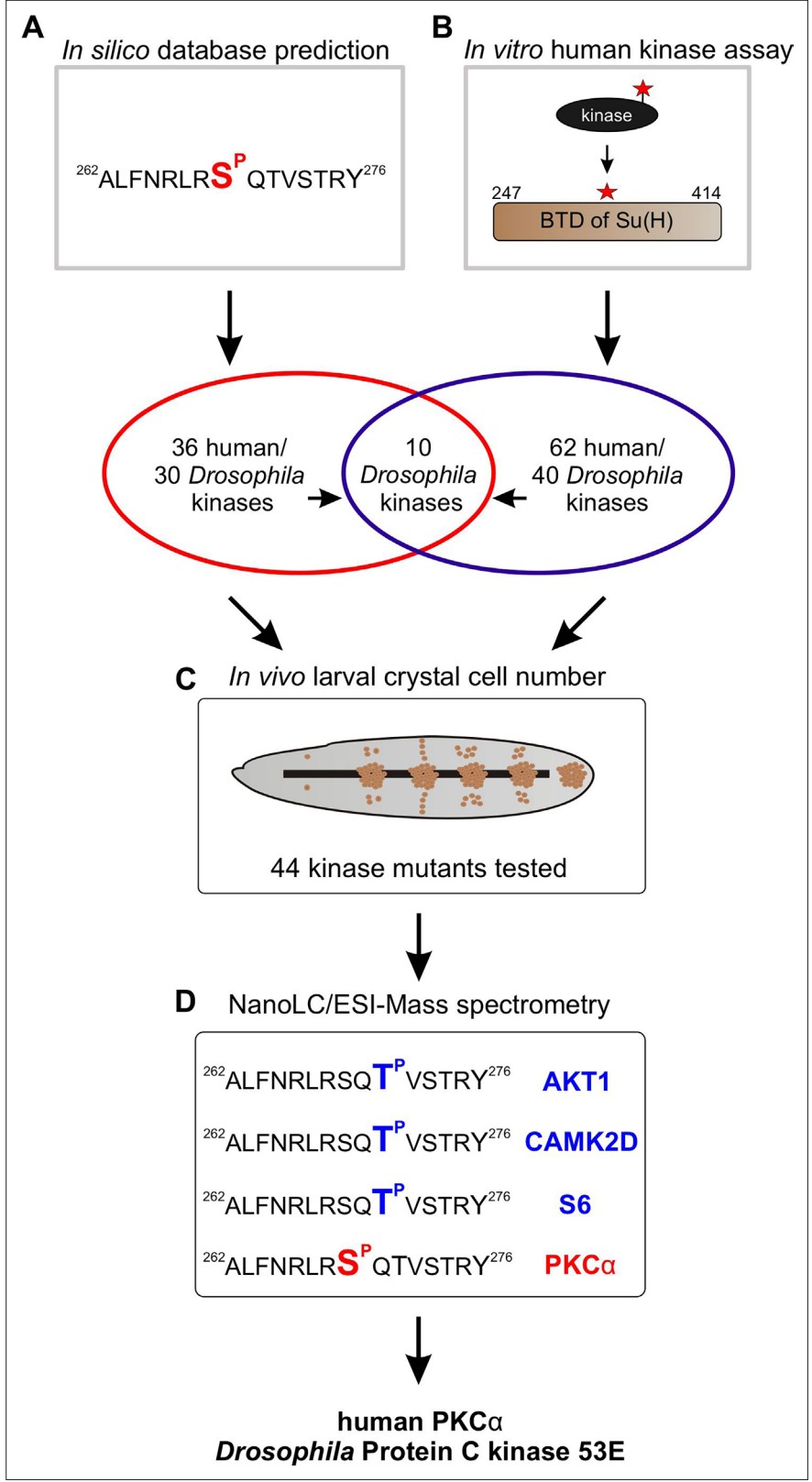

**Figure 3.** Pipeline of screening procedures for kinase candidates triggering phosphorylation at Su(H)$^{S269}$. (**A**) In silico screening of database(s) predicting kinase recognition motif in Su(H)$^{S269}$; see *Supplementary file 1*. (**B**) In vitro assay screening 245 human Ser/Thr kinases for their ability to phosphorylate the beta-trefoil domain (BTD) domain of Su(H); see *Supplementary file 2*. (**C**) In vivo screen of 44 different *Drosophila* kinase mutants for

*Figure 3 continued on next page*

*Figure 3 continued*

crystal cell occurrence in third instar larvae; see **Supplementary file 3**, **Supplementary file 4**, and **Figure 3— figure supplement 1**. (**D**) NanoLC/ESI mass spectrometry with active human kinases monitoring their ability to phosphorylate the given Su(H) peptide. PKCα phosphorylates S269, whereas AKT1, CAMK2D, and S6 kinase prefer T271. Spectra are shown in **Figure 3—figure supplement 2**.

The online version of this article includes the following figure supplement(s) for figure 3:

**Figure supplement 1.** Overview of the results from the kinase screen.

**Figure supplement 2.** MS/MS spectra of the phosphorylated Su(H) peptide.

the domain (**Kovall and Blacklow, 2010**), and to reduce the number of potential phospho-sites at the same time present in full-length Su(H). With this approach, we ended up with 62 human Ser/Thr kinases (25% of the tested kinases) that use the BTD of Su(H) as an in vitro substrate, corresponding to 40 different kinases in *Drosophila* (**Supplementary file 2**). Both sets of data, biochemical and bioinformatics, were used to generate a list of 44 candidates to be analysed by genetic means. The candidates were further screened for an imbalanced hematopoiesis. We reasoned that mutants affecting a relevant kinase gene involved in the phosphorylation of Su(H) should display increased crystal cell numbers similar to what was observed in the *Su(H)*[S269A] mutant (**Frankenreiter et al., 2021**). Larvae of 13 different kinase mutants, and the progeny of 44 UAS-RNAi and/or UAS-kinase dead transgenes crossed with *hml*-Gal4, a blood cell-specific Gal4 driver line, were tested (**Supplementary file 3**). To this end, the larvae were subjected to heating for a visualisation and quantification of sessile crystal cells (**Rizki, 1957**; **Lanot et al., 2001**). About a third of the kinase mutants were similar to the control, whereas mutations in four kinases impeded crystal cell development or prevented it altogether. Unexpectedly, the majority of the tested kinase mutants exhibited elevated crystal cell numbers, however to a different degree (**Figure 3—figure supplement 1**, **Supplementary file 4**; **Deichsel et al., 2024**). Nineteen kinase mutants matched closely the *Su(H)*[S269A] phenotype, making those the most promising candidates to being involved in the phosphorylation of Su(H) at S[269]. Six of those were within the cluster of 10 candidates singled out by the in silico and the in vitro screens (**Figure 3A and B**). Using commercially available, activated human kinases, we were able to test five candidates, AKT1, CAMK2D, GSK3B, S6, and PKCα in vitro by MS/MS analysis on the Su(H) peptide ALFNRLR**S**[8]QTVSTRY, where Serine 8 (S8) corresponds to S269 in Su(H). Only PKCα unambiguously phosphorylated the given peptide at Serine 8. Whereas GSK3B did not phosphorylate the peptide at all, AKT1, CAMK2D, and S6 piloted Threonine 10, corresponding to Threonine 271 in Su(H) (**Figure 3D**, **Figure 3—figure supplement 2**). Together, these data support the idea that PKCα corresponding to Pkc53E in *Drosophila* is part of the kinase network mediating the phosphorylation of Su(H) at S269.

## Role of Pkc53E in the phosphorylation of Su(H)[S269]

Confirming the MS/MS data, human PKCα was able to phosphorylate the respective Su(H) peptide (S[wt]) similar to its defined pseudosubstrate PS (**Kochs et al., 1993**), whereas the S8A mutant peptide (S[SA]) was accepted only half as well in an ADP-Glo assay, indicating that S269 is a preferred substrate (**Figure 4A**). Bacterially expressed and purified *Drosophila* Pkc53E, however, did neither accept the PS nor the Su(H) peptides (**Figure 4B**). Pkc53E activity, however, was stimulated by the agonistic phorbol ester PMA (phorbol 12-myristate 13-acetate) (**Blumberg et al., 1983**; **Nakashima, 2002**) to phosphorylate the PS and Su(H) peptide S[wt] but not the S8A mutant peptide S[SA] (**Figure 4C**). To generate an activated form of Pkc53E, we exchanged four codons by in vitro mutagenesis, three (T508D, T650D, and S669D) mimicking phosphorylation in the kinase and C-terminal domains, respectively, and one in the pseudosubstrate domain (A34E) (**Figure 4—figure supplement 1**; **Gould and Newton, 2008**). The resultant Pkc53E[EDDD] kinase accepted the PS, and the Su(H) peptide S[wt] even better, but not the S8A mutant peptide S[SA], demonstrating the specificity of the phosphorylation (**Figure 4D**). As predicted for a fully activated kinase, PMA was unable to boost Pkc53E[EDDD] protein activity any further (**Figure 4E**). The bacterially expressed Pkc53E[EDDD] kinase, however, was a magnitude less active compared to commercial PKCα, perhaps reflecting poor quality of the bacterially expressed protein, or indeed an intrinsic biochemical property of the *Drosophila* enzyme.

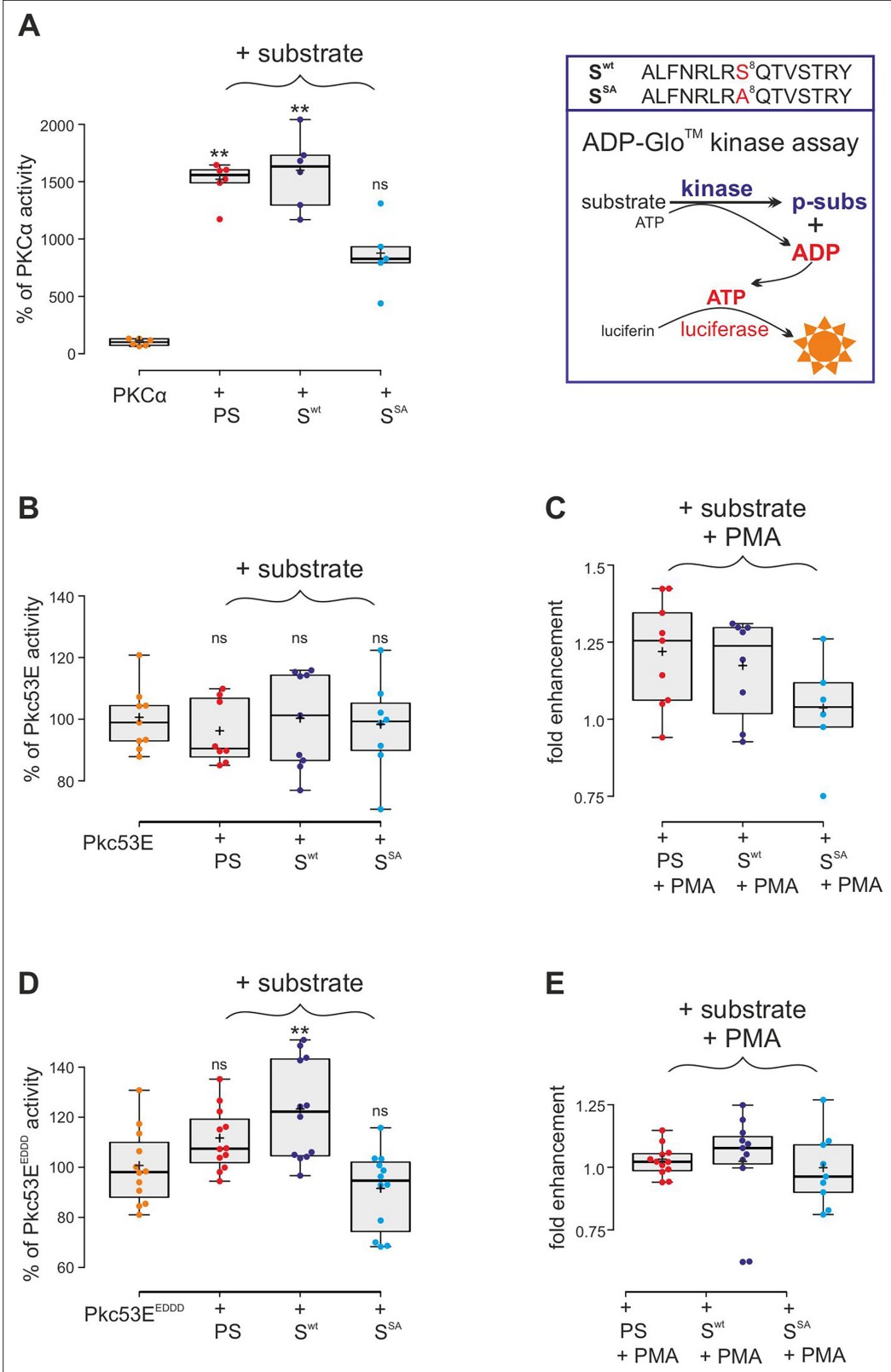

**Figure 4.** Kinase assays using activated PKCα and *Drosophila* Pkc53E variants. (**A**) Right, schema of ADP-Glo assay to quantify kinase activity. The wild-type (S^wt) and mutant (S^SA) Su(H) peptides 262–276 offered as specific kinase substrates are indicated above. Left, the commercially available, active human PKCα very efficiently phosphorylates the pseudosubstrate PS and the Su(H) S^wt peptide, but less efficiently the S^SA mutant peptide. Activity is given as

*Figure 4 continued on next page*

*Figure 4 continued*

percentage of the auto-active kinase without substrate. Each dot represents one experiment (n=5-6). (**B**) Bacterially expressed Pkc53E has no activity on any of the offered substrates PS, S$^{wt}$, or S$^{SA}$ (n=8-9). (**C**) PMA (phorbol 12-myristate 13-acetate) raised Pkc53E activity to nearly 125% for PS and S$^{wt}$ but not for S$^{SA}$ (n=6-9). (**D**) Activated Pkc53E$^{EDDD}$ phosphorylates PS and S$^{wt}$ but not for S$^{SA}$ (n=12). (**E**) Addition of PMA does not change Pkc53E$^{EDDD}$ activity (n=9-12). Statistical analyses were performed with Kruskal-Wallis test followed by Dunn's test (**A, C, E**) or ANOVA followed by Tukey's approach in (**B, D**) with **p<0.01, ns (not significant p≥0.05).

The online version of this article includes the following source data and figure supplement(s) for figure 4:

**Source data 1.** Raw data and statistical analysis.

**Figure supplement 1.** Conservation of Pkc53E and generation of an activated Pkc53E$^{EDDD}$ isoform.

## The PKC-agonist PMA influences Su(H) activity and blood cell homeostasis

Our data so far indicated that Su(H) is a phospho-target of Pkc53E which reduces its activity by affecting its DNA binding. In this case, we might expect an influence of the general PKC activator PMA on both, Su(H) activity and Notch-mediated crystal cell formation. We tested the former in a *RBPJ$^{ko}$* HeLa cell system (*Wolf et al., 2019*), measuring Notch reporter gene activation by Su(H)-VP16. To this end, a Su(H)-VP16 gene fusion was cloned under HSV-TK promoter control, which is unresponsive to PMA in HeLa cells (*Shifera and Hardin, 2009*). Su(H)-VP16 protein is independent of Notch activity itself, allowing to directly monitor the influence of PMA on Su(H) activity. Indeed, Su(H)-VP16's ability to activate reporter gene transcription was reduced by more than half in the presence of PMA (*Figure 5A*). This is in agreement with a PMA-mediated activation of endogenous PKC in the transfected HeLa cells, resulting in Su(H)-VP16 phosphorylation, loss of DNA-binding activity, and reduced transcriptional activation. Accordingly, repression could be reverted by the addition of kinase-inhibitor staurosporine (STAU) (*Karaman et al., 2008*). In fact, STAU alone already increased Su(H)-VP16 transcriptional activity, suggesting that inhibitory phosphorylation of Su(H) occurs in HeLa cells (*Figure 5A*). Expression levels of Su(H)-VP16, however, remained unchanged by the treatments (*Figure 5—figure supplement 1*).

Next, we assayed the effect of PMA on larval crystal cell formation. If, as expected, PMA increased Pkc53E activity, Su(H) should be inactivated by phosphorylation with decreased crystal cell numbers as a consequence. We fed PMA to *Drosophila* larvae and assayed the numbers of sessile crystal cells. In agreement with our expectations, crystal cell numbers dropped very strongly, suggesting efficient phosphorylation and inactivation of Su(H) protein by PKCs (*Figure 5B*). As predicted by the above experiments, this effect was alleviated by STAU. Owing to the global inhibition of kinases by STAU, however, a rise in crystal cells was expected, because many kinases restrict their numbers (see *Figure 3—figure supplement 1*; *Deichsel et al., 2024*). In contrast, *Su(H)$^{S269A}$* mutant larvae displayed an increased number of crystal cells which can be attributed to the fact that here, Su(H) can no longer be phosphorylated and hence, is overactive in this context. Feeding PMA to *Su(H)$^{S269A}$* larvae caused only a minor drop of excessive crystal cell numbers, which could be due to other kinases acting on *Su(H)$^{S269}$* or due to other Pkc53E substrates involved in crystal cell development. Again, kinase inhibition by STAU largely reversed this effect (*Figure 5B*). In conclusion these data support the notion that Su(H) activity is regulated in vitro and in vivo by PKC activity in the context of blood cell homeostasis.

## Pkc53E is required for normal blood cell homeostasis in *Drosophila* larvae

*Su(H)$^{S269A}$* mutant larvae develop an excess of crystal cells, both in the hemolymph and in the lymph glands, due to a failure to downregulate respective Notch activity in the particular progenitor cells via the phosphorylation of Su(H) protein (*Frankenreiter et al., 2021*) (see *Figure 1A and B*). Assuming Pkc53E has a major role in the phosphorylation of Su(H), we should expect a similar phenotype in a *Pkc53E* mutant due to the inability to phosphorylate any substrate. In order to test this assumption directly, we assessed the number of crystal cells in larval lymph glands as well as in sessile crystal cells in several loss-of-function backgrounds of *Pkc53E*. To this end, we used the *Pkc53E$^{Δ28}$* allele, which by RT-PCR is a null mutant (*Figure 6—figure supplement 1*). In addition, we used two different RNAi-lines and one sgRNA line under UAS-control to knock down *Pkc53E* activity specifically in progenitor

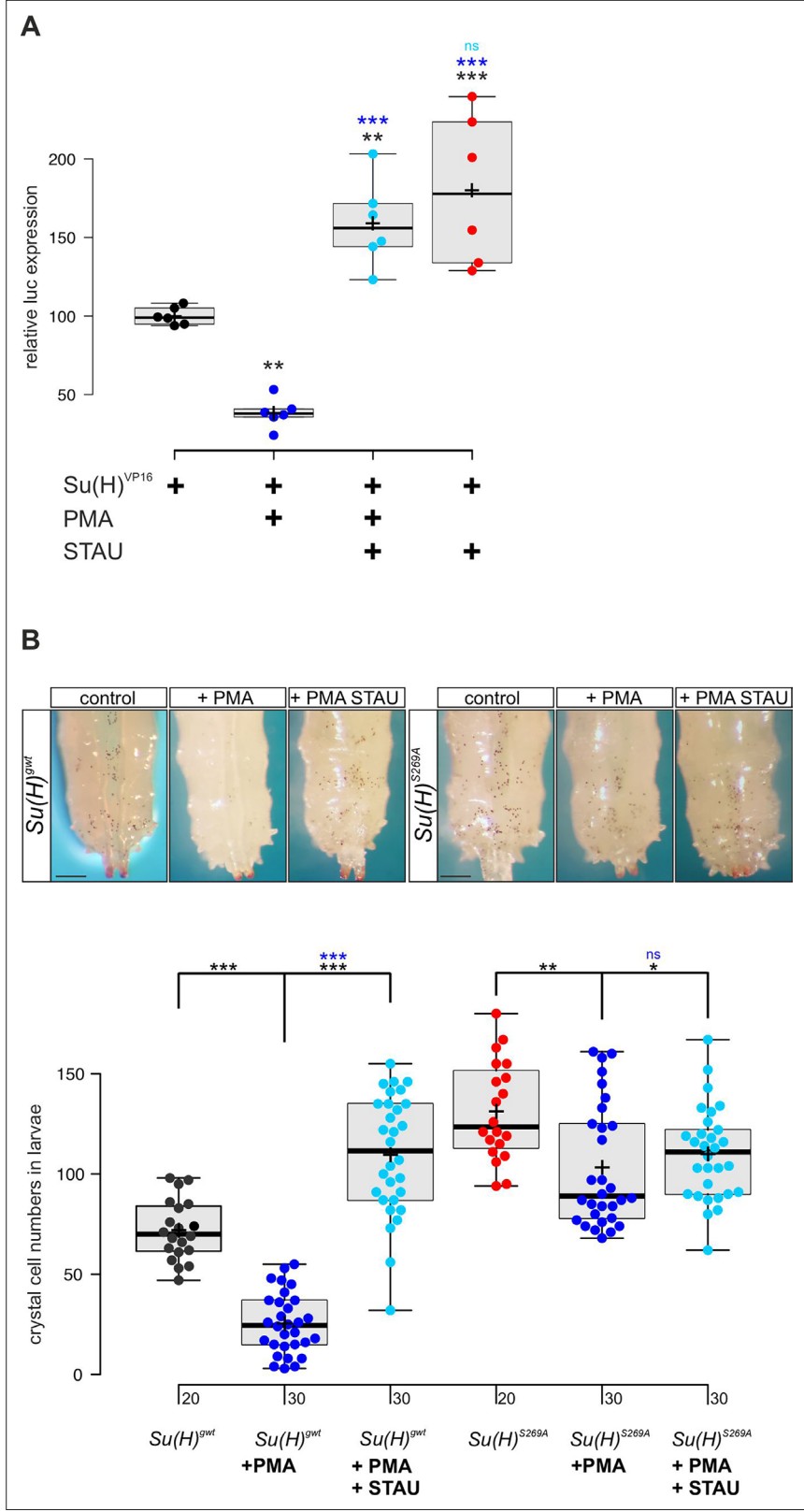

**Figure 5.** PMA (phorbol 12-myristate 13-acetate) inhibits Su(H) transcriptional activity in vitro and crystal cell formation in vivo. (**A**) Expression of NRE-luciferase reporter gene in *RBPj*[jko] HeLa cells, transfected with 2xMyc-Su(H)-VP16 [Su(H)[VP16]]. Luciferase activity is given relative to the reporter construct normalised to Su(H)-VP16 set to 100%. Addition of PMA causes reduction of Su(H)-VP16-dependent transcriptional activity to about 40%, which is

*Figure 5 continued on next page*

*Figure 5 continued*

reversed by the kinase inhibitor staurosporine (STAU). STAU itself results in increased Su(H)-VP16 activity. Each dot represents one experiment (n=6). Statistical analysis was performed with ANOVA followed by Dunnet's multiple comparison test with ***p<0.001, **p<0.01 relative to Su(H)-VP16 alone (black asterisks) or to Su(H)-VP16 plus PMA (blue asterisks). (**B**) Number of melanised larval crystal cells determined in the last two segments of larvae fed with fly food plus 10% DMSO (control), or with fly food supplemented with 1 mM PMA, or 1 mM PMA plus 0.2 mM STAU (n=20 or n=30, as indicated). Note strong drop of crystal cell numbers in the *Su(H)^gwt* control fed with PMA, and a reversal by STAU addition even above control levels. In contrast, PMA has a small effect on crystal cell number in the *Su(H)^S269A* mutant, which is reversed by STAU. Representative animals are shown above; scale bar, 250 μm. Statistical analysis was performed by ANOVA followed by Tukey's multiple comparison test (***p<0.001), significant differences are colour coded.

The online version of this article includes the following source data and figure supplement(s) for figure 5:

**Source data 1.** Raw data and statistical analysis.

**Figure supplement 1.** Expression of Su(H)^VP16-myc is not influenced by PMA (phorbol 12-myristate 13-acetate) or staurosporine (STAU).

**Figure supplement 1—source data 1.** Original western blot.

**Figure supplement 1—source data 2.** Original western blot, labelled.

cells within the developing lymph gland using *lz*-Gal4 and in the hemolymph using *hml*-Gal4, respectively (**Lebestky et al., 2000**). As the *hml*-Gal4 driver is active in plasmatocytes and pre-crystal cells (**Mukherjee et al., 2011**; **Tattikota et al., 2020**), it should affect *Pkc53E* activity prior to crystal cell commitment in the hemolymph. However, within the lymph gland, *hml* appears specific to the plasmatocyte lineage and not present in crystal cell precursors. Instead, we choose *lz*-Gal4 for the *Pkc53E* knockdown, as *lz* is expressed in differentiating crystal cells of the lymph gland (**Lebestky et al., 2000**; **Blanco-Obregon et al., 2020**).

In any context tested, the number of crystal cells was strongly increased matching those of the *Su(H)^S269A* mutant (**Figure 6**, **Figure 6—figure supplement 2**). The similar phenotypes imply that Pkc53E acts through the phosphorylation of Su(H) specifically within hemocytes to restrict crystal cell differentiation. However, as outlined above, the majority of kinase mutants displayed increased crystal cell numbers, raising the possibility of a fortuitous accordance. If the increase of crystal cell numbers in *Pkc53E* mutants is independent of Su(H) phosphorylation, we should expect an additive effect if we combine the two mutants. The double mutants *Su(H)^S269A Pkc53E^Δ28* were generated by genetic recombination; they displayed the same range of excessive crystal cell numbers as the single mutants (**Figure 7A and B**, **Figure 7—figure supplement 1**). Moreover, the strongly reduced number of crystal cells observed in the *Su(H)^S269D* mutant was not increased by *Pkc53E^Δ28* (**Figure 7A and B**), indicating that Pkc53E indeed acts upstream of Su(H), or directly on Su(H). If the latter is the case, we may expect the two proteins to form complexes in vivo. Indeed, we could co-precipitate Su(H)-Pkc53E protein complexes, both from *Drosophila* heads containing hemocytes (**Sanchez Bosch et al., 2019**), and from the larval hemolymph (**Figure 7C and D**). Specific co-precipitation was eased by using fly strains expressing m-Cherry-tagged Su(H) (**Praxenthaler et al., 2017**) and HA-tagged Pkc53E, respectively. Together, these data demonstrate that *Pkc53E* has an important role in blood cell homeostasis that can be largely explained by its activity to phosphorylate Su(H), thereby regulating the activity of Notch target genes during hemocyte and lymph gland development.

## Pkc53E mutants are immune-compromised

According to our hypothesis, Pkc53E phosphorylates Su(H) in response to an immune challenge by parasitic wasp infestation. In fact, a Pkc53E-eGFP fusion protein expressed from the *Pkc53E* locus via protein trap (**Lee et al., 2018**) was observed in the cytoplasm of all blood cell types independent of wasp infestation (**Figure 8**). Moreover, *Pkc53E* mRNA was expressed in cells of the larval hemolymph (**Figure 8—figure supplement 1**). Hence, we would expect that a loss of Pkc53E function should affect the ability of *Drosophila* larvae to fight wasp infestations similarly to the *Su(H)^S269A* mutant. This was indeed the case. Firstly, the *Pkc53E^Δ28* null mutant was likewise impaired in fighting an infestation with the wasp *A. japonica* as was the *Su(H)^S269A* mutant or the double mutant *Su(H)^S269A Pkc53E^Δ28* with only about 4% surviving flies versus 14% in the control (**Figure 9A**). In contrast to *Su(H)^S269A*, however, the *Pkc53E^Δ28* mutant larvae contained significantly lower hemocyte numbers independent

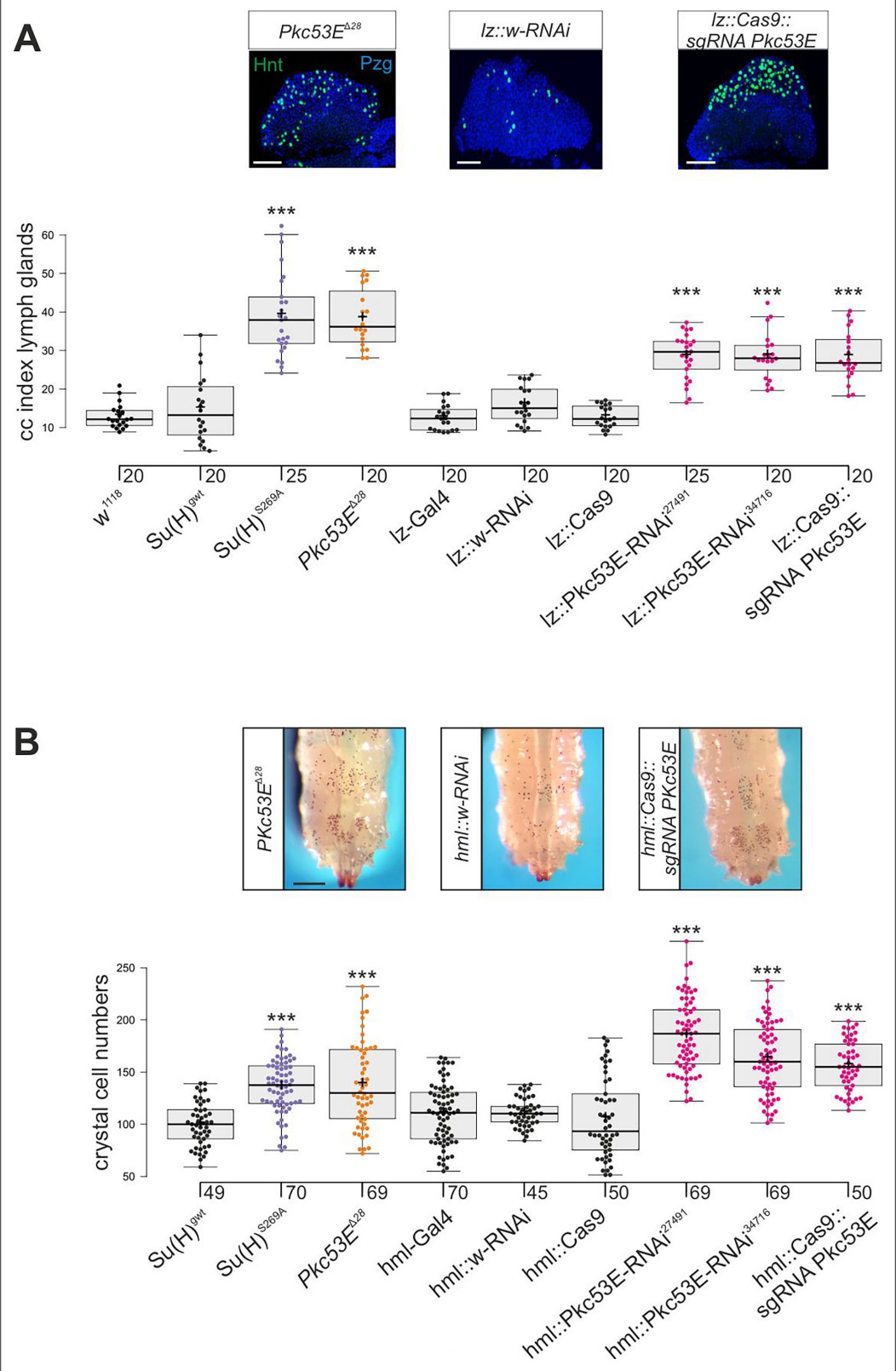

**Figure 6.** Loss of Pkc53E causes a gain of crystal cell number. Depletion of *Pkc53E* activity in the *Pkc53E^Δ28* mutant or after knockdown by *Pkc53E*-RNAi or sg*Pkc53E* with the help of the Gal4-UAS system using *lz*-Gal4 (**A**) or *hml*-Gal4 (**B**). Controls as indicated. (**A**) Crystal cell index in lymph glands; each dot represents the value of an analysed lobus (n=20 or n=25, as indicated). Examples of *Pkc53E^Δ28*, *lz::Pkc53-RNAi,* and *lz::Cas9 sgPkc53E* are shown above.

*Figure 6 continued on next page*

*Figure 6 continued*

Crystal cells are labelled with Hnt (green), the lobe is stained with α-Pzg (blue). Scale bar, 50 µm. Representative images of lymph glands for each genotype are shown in *Figure 6—figure supplement 2A*. Statistical analysis by ANOVA followed by Tukey's multiple comparison test relative to controls with ***p<0.001. (**B**) Melanised crystal cells enumerated from the last two segments of larvae with the given genotype (n=45–70 as indicated). Examples of respective *Pkc53E*$^{Δ28}$, *lz::Pkc53*-RNAi, and *lz::Cas9 sgPkc53E* larvae are shown above. Scale bar, 250 µm. Representative larval images for each genotype are shown in *Figure 6—figure supplement 2B*. Statistical analysis by Kruskal-Wallis test, followed by Dunn's test relative to controls with ***p<0.001. Note that there were no significant differences between any of the controls shown in black.

The online version of this article includes the following source data and figure supplement(s) for figure 6:

**Source data 1.** Raw data and statistical analysis.

**Figure supplement 1.** *The Pkc53E*$^{Δ28}$ *allele is a null mutant.*

**Figure supplement 1—source data 1.** Original agarose gel showing RT-PCR of Pkc53E$^{Δ28}$ mutant including relevant controls.

**Figure supplement 1—source data 2.** Original agarose gel showing RT-PCR of Pkc53E$^{Δ28}$ mutant including relevant controls - labelled.

**Figure supplement 2.** Representative images for the various settings.

of infestation, perhaps partly explaining the poor immune response (*Figure 9—figure supplement 1*; *McGonigle et al., 2017*). Without infestation, however, *Pkc53E*$^{Δ28}$ mutant larvae developed normally to adulthood (*Figure 9—figure supplement 1*). Moreover, when *Pkc53E*$^{Δ28}$ was infested with the parasitic wasp *L. boulardi*, lamellocyte numbers in the hemolymph did not reach wild-type levels, and they were almost absent from the larval lymph glands (*Figure 9B and C*). Apparently, the *Pkc53E*$^{Δ28}$ null mutant is impaired in recognising parasitic wasp infestation or is hampered responding to it, e.g., by the phosphorylation of Su(H), demonstrating the involvement of this kinase in the immune response of *Drosophila* to parasitoid wasp infestation.

## Discussion

The Notch pathway is highly conserved between invertebrates and vertebrates, with regard to both the underlying molecular principles and the biological processes it is involved, including hematopoiesis and immune defense. During mammalian hematopoiesis, Notch plays a fundamental role in stem cell maintenance and proliferation as well as in the differentiation of blood cell precursors, notably in T-cell development. Accordingly, aberrant Notch signalling activity has profound consequences for blood cell homeostasis that may result in leukemia (reviewed in *Radtke et al., 2010*; *Siebel and Lendahl, 2017*; *Banerjee et al., 2019*; *Gallenstein et al., 2023*). Hence, the principles of the regulation of Notch signalling activity are of great interest.

Notch signals are transduced by CSL-type DNA-binding proteins that are pivotal to Notch pathway activity, as no signal transduction could occur in their absence or in the instance of a lack of DNA binding. CSL proteins are extremely well conserved. They act as a molecular switch, either activating or repressing Notch target genes, depending on the recruited cofactors. Upon ligand binding, the Notch receptor is cleaved, and the biologically active Notch intracellular domain assembles an activator complex with CSL and further cofactors. In the absence of Notch receptor activation, however, CSL recruits co-repressors for gene silencing; in *Drosophila* repressor complex formation is mediated by Hairless (reviewed in *Maier, 2006*; *Borggrefe and Oswald, 2009*; *Kovall and Blacklow, 2010*; *Bray, 2016*; *Giaimo et al., 2021*). Accordingly, a loss of DNA binding by CSL is expected to affect both, the activation of Notch targets in the presence and their repression in the absence of Notch signals. This dual effect was observed in the context of wing development in cells homozygous for the DNA-binding defective *Su(H)*$^{S269D}$ variant: a failure of Notch target gene expression in areas of high Notch activity, as well as a de-repression of Notch target genes in areas outside (*Frankenreiter et al., 2021*). Phosphorylation of Su(H) hence entails not only the inhibition of Notch activity, but likewise a de-regulation of genes silenced by Su(H)-repressor complexes. Previously, we have shown that the regulation of Notch activity during hemocyte differentiation is independent of Hairless, but rather relies on the phosphorylation of Su(H) (*Frankenreiter et al., 2021*). Moreover, Notch activity needs to be downregulated before lamellocyte formation during parasitism, arguing for an inhibition of Notch

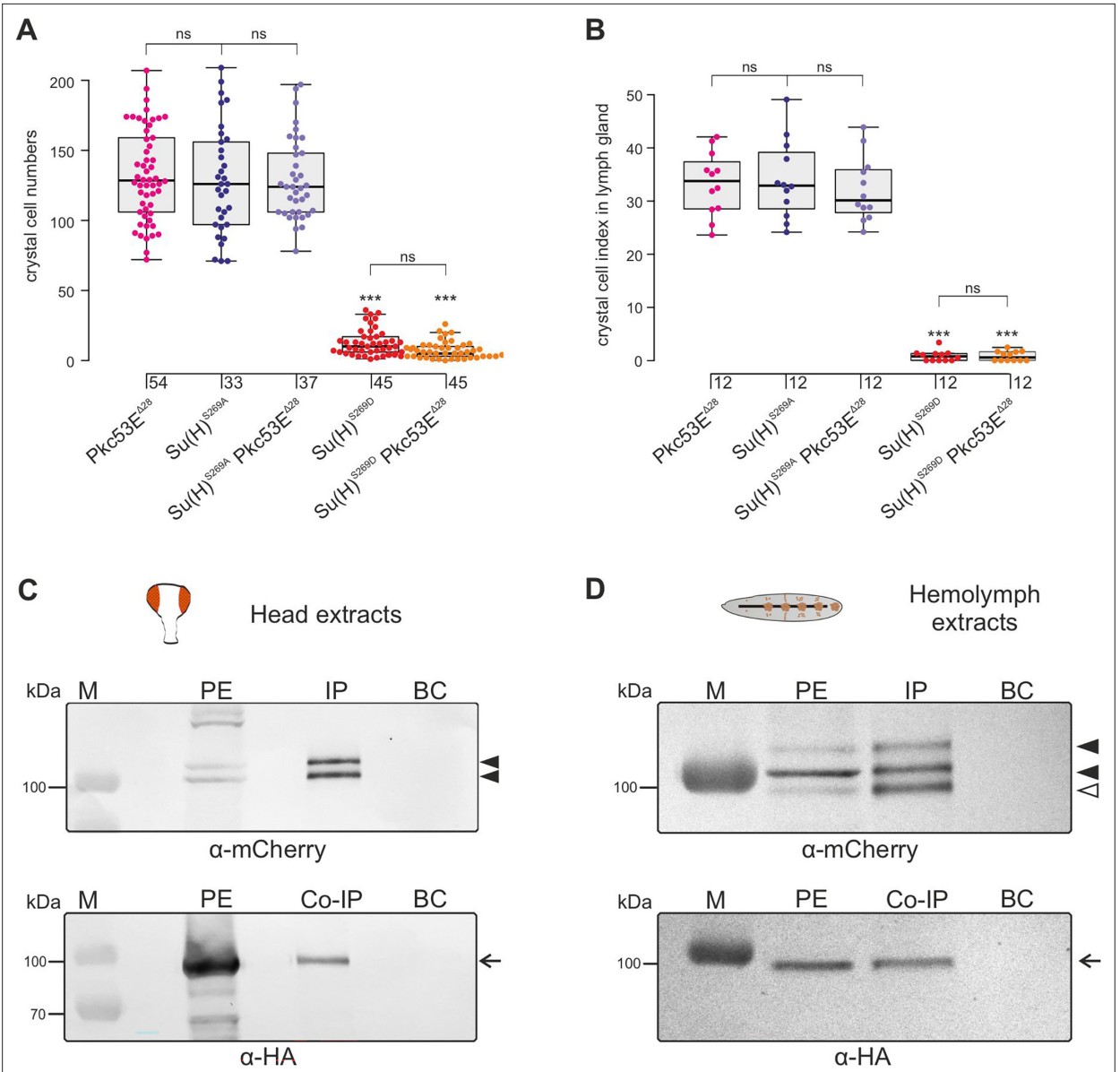

**Figure 7.** Pkc53E interacts with Su(H) at a genetic and a physiological level. (**A**) Larval crystal cell numbers and (**B**) crystal cell indices in lymph glands were determined in the given genotypes. Each dot represents one analysed larva (n, as indicated) (**A**) or lymph gland lobus (n=12). (**B**) Statistical analysis by Kruskal-Wallis test, followed by Dunn's test relative to controls with ***p<0.001; p≥0.05 ns (not significant). Representative images of sessile crystal cells and of lymph glands for each genotype are shown in *Figure 7—figure supplement 1*. (**C, D**) Co-immunoprecipitation of Pkc53E[HA] with Su(H)[gwt-mCh] protein. RFP-Trap IP was performed with protein extracts from 400 heads (**C**) or 25 third instar larvae (**D**), respectively. UAS-Pkc53E-HA expression was induced with *Gmr*-Gal4 in the head or with *hml*-Gal4 in the hemolymph. Endogenous mCherry-tagged Su(H) was trapped and detected with anti-mCherry antibodies (black arrowheads). The lowest band from the hemolymph is presumably a degradation product (open arrowhead in (**D**)). HA-tagged Pkc53E was specifically co-precipitated as detected with anti-HA antibodies (arrow). 10% of the protein extract (PE) used for the IP-Trap was loaded for comparison. BC corresponds to the Trap with only agarose beads as a control. M, prestained protein ladder; protein size is given in kDa.

The online version of this article includes the following source data and figure supplement(s) for figure 7:

**Source data 1.** Original, uncropped western blots of Su(H)-mCh and Pkc53E-HA co-IP in head extracts and hemolymph, respectively, shown in *Figure 7C and D*.

**Source data 2.** Original, uncropped western blots of Su(H)-mCh and Pkc53E-HA co-immunoprecipitation (co-IP) in head extracts and hemolymph, respectively, shown in *Figure 7C and D* - labelled.

**Source data 3.** Raw data and statistical analysis.

**Figure supplement 1.** Representative images for the various settings.

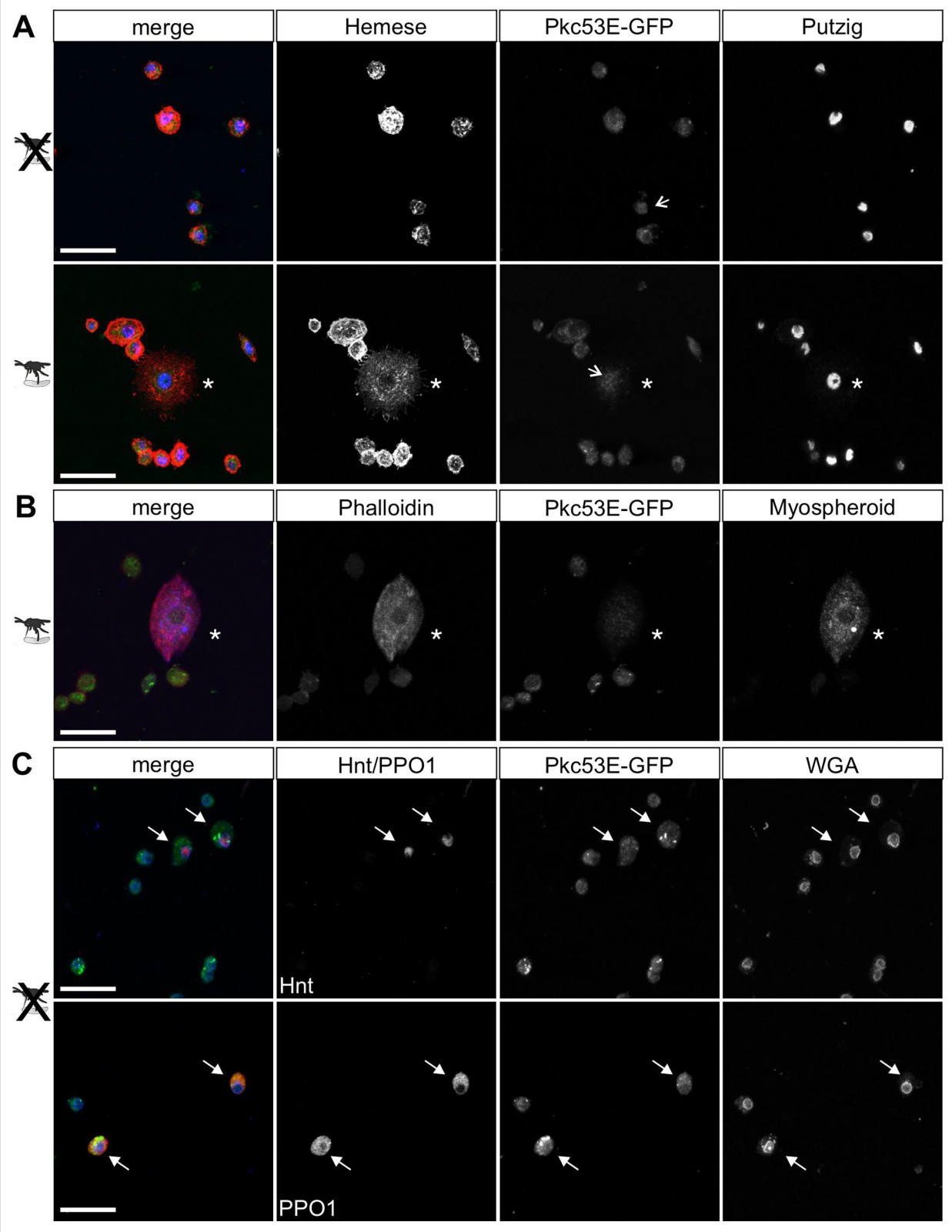

**Figure 8.** Pkc53E-eGFP is expressed in the cytoplasm of all hemocytes. Hemocytes derived from Pkc53E-eGFP expressing larvae, either infested or not infested with *L. boulardi* were stained with the antibodies and compounds indicated. (**A**) Pkc53E-eGFP is present in the cytoplasm of hemocytes independent of wasp infection. Note complete overlap with Hemese (red) marking all types of blood cells; Putzig (blue) labels nuclei. Asterisk denotes lamellocyte in hemolymph of infected larvae. Arrows point to nuclei expressing Pkc53E-eGFP. (**B**) Pkc53E-eGFP is expressed in the cytoplasm of a

*Figure 8 continued on next page*

Figure 8 continued

lamellocyte (asterisk), labelled with myospheroid (blue) and rhodamine-coupled phalloidin (red). (C) Pkc53E-eGFP is enriched in the cytoplasm of crystal cells (arrow), labelled either with Hnt (red) or PPO1 (red), as indicated. Wheat germ agglutinin (WGA, blue) served to label nuclear lamina. Scale bar, 25 μm.

The online version of this article includes the following source data and figure supplement(s) for figure 8:

Figure supplement 1. Pkc53E is expressed in hemocytes.

Figure supplement 1—source data 1. Original agarose gel showing RT-PCR for Pkc53E expression in hemocytes.

Figure supplement 1—source data 2. Original agarose gel showing RT-PCR for Pkc53E expression in hemocytes - labelled.

activity rather than a de-repression of Notch target genes resulting from Su(H) phosphorylation during wasp parasitism.

Studies of *Drosophila* hematopoiesis have uncovered a pleiotropic role of Notch in all the hematopoietic compartments, where Notch activity needs to be precisely regulated to ensure blood cell homeostasis (reviewed in *Banerjee et al., 2019*; *Csordás et al., 2021*). During blood cell formation, Notch directs the crystal cell lineage. Accordingly, a downregulation of Notch activity causes a loss of crystal cells, whereas a gain of Notch activity results in increased numbers (*Duvic et al., 2002*; *Lebestky et al., 2003*; *Terriente-Felix et al., 2013*; *Ghosh et al., 2015*; *Frankenreiter et al., 2021*). In our earlier work, we have shown that the phosphorylation at S269 in the BTD of Su(H) impairs DNA-binding activity, and hence the capability of Su(H) to act as a transcriptional regulator in the context of blood cell development, without affecting Su(H) protein expression (*Nagel et al., 2017*; *Frankenreiter et al., 2021*). Accordingly, phospho-deficient $Su(H)^{S269A}$ mutants develop an excess of crystal cells. Now we provide evidence that Su(H) phosphorylation is likewise involved in parasitoid wasp defense, pointing to a dual use of the Notch pathway in blood cell homeostasis as well as in stress response, which is typical for the myeloid system (*Banerjee et al., 2019*).

The primary cellular immune response of *Drosophila* to fight parasitoid wasp infestation is an encapsulation of the parasite egg to terminate its further development. Albeit metabolically extremely costly, *Drosophila* larvae have to remodel their entire hematopoietic system to generate the masses of lamellocytes required for the encapsulation (reviewed in *Letourneau et al., 2016*; *Kim-Jo et al., 2019*; *Csordás et al., 2021*; *Hultmark and Andó, 2022*). There are two major sources for the lamellocytes. One is the transdifferentiation of circulating plasmatocytes, released from the sessile compartment and proliferating upon immune challenge (*Márkus et al., 2009*; *Honti et al., 2010*; *Stofanko et al., 2010*; *Vanha-Aho et al., 2015*; *Anderl et al., 2016*). Second is a massive expansion of prohemocytes in the lymph gland followed by a differentiation to lamellocytes and their release due to the premature disintegration of the gland (*Lanot et al., 2001*; *Sorrentino et al., 2002*; *Louradour et al., 2017*; *Cho et al., 2020*). Wasp infestation hence provokes a biased commitment to the lamellocyte lineage at the expense of crystal cells, as the two lineages are mutually exclusive (*Krzemien et al., 2010*; *Tattikota et al., 2020*; *Cho et al., 2020*; *Cattenoz et al., 2020*).

The processes underlying the *Drosophila* immune response to wasp parasitism are well understood, sharing many molecular details with the inflammatory responses of vertebrates (reviewed in *Letourneau et al., 2016*; *Kim-Jo et al., 2019*; *Banerjee et al., 2019*; *Kharrat et al., 2022*). Interestingly, sterile injury of *Drosophila* larvae initiates a likewise immune response covering all aspects of wasp-mediated immune challenge (*Evans et al., 2022*). The epidermal penetration by the wasp ovipositor causes a burst of hydrogen peroxide at the injury site via the activation of the NADPH oxidase DUOX. The oxidative stress then induces a systemic activation of Toll/NF-κB and JNK signalling within circulating hemocytes as well as within the cells of the posterior signalling centre of the lymph gland. Following cytokine release, JAK/STAT and EGFR signalling pathways are activated non-cell autonomously driving the (trans-)differentiation of (pro-)hemocytes to lamellocyte fate and lymph gland dispersal (*Nappi et al., 1995*; *Schlenke et al., 2007*; *Sinenko et al., 2011*; *Gueguen et al., 2013*; *Razzell et al., 2013*; *Louradour et al., 2017*; *Chakrabarti and Visweswariah, 2020*; *Evans et al., 2022*; reviewed in *Letourneau et al., 2016*; *Banerjee et al., 2019*). Whereas the switch to lamellocyte fate is well elaborated, less is known about the processes underlying the simultaneous suppression of crystal cell fate. Obviously, this step requires a downregulation of Notch signalling activity, as the Notch pathway is instrumental to crystal cell fate in all the hematopoietic compartments (*Duvic et al., 2002*; *Lebestky et al., 2003*; *Schlenke et al., 2007*; *Small et al., 2014*; reviewed in *Banerjee*

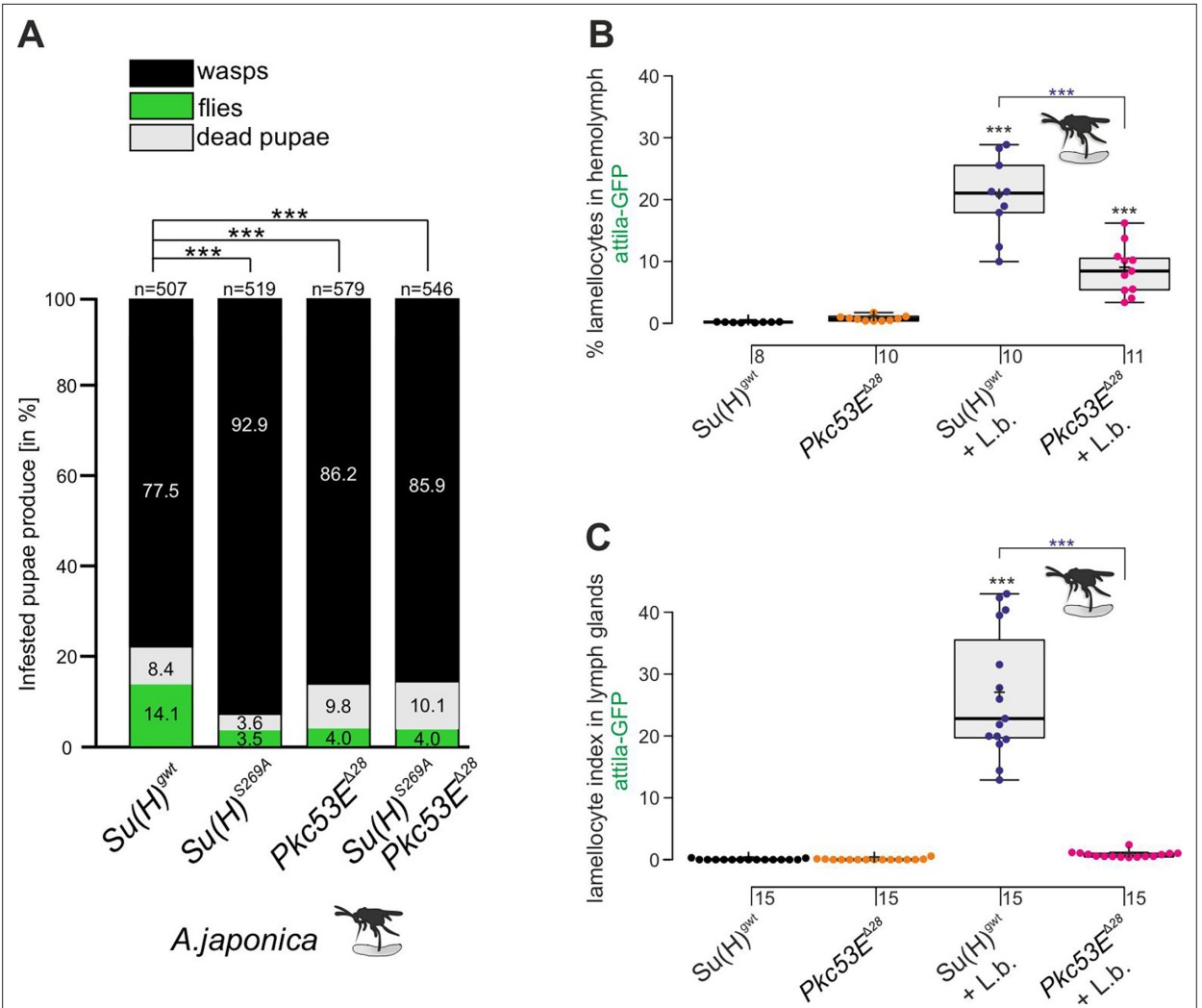

**Figure 9.** The Pkc53E[Δ28] null mutant is immune-compromised. (**A**) Resistance of *Pkc53E[Δ28]* and the double mutant *Su(H)[S269A] Pkc53E[Δ28]*, compared to *Su(H)[gwt]* and *Su(H)[S269A]* for control, to the infestation with parasitic wasp strain *A. japonica*. Numbers of eclosed wasps versus flies as well as of dead pupae are presented in relation to the total of infested pupae (n, number of infested pupae). Statistical analysis by ANOVA followed by Tukey's multiple comparison test relative to control *Su(H)[gwt]*; *p<0.05; ***p<0.001. (**B, C**) Quantification of lamellocytes labelled with the *atilla*-GFP reporter in the circulating hemolymph (**B**) or in the lymph glands (**C**), in uninfested conditions or upon wasp infestation as indicated. Representative images of hemolymph and of lymph glands for each genotype are shown in **Figure 9—figure supplement 2**. (**B**) Fraction of GFP-positive lamellocytes relative to the total of DAPI-stained hemocytes in the pooled hemolymph from 10 larvae. Each dot represents one larval pool (n, number of experiments as shown). (**C**) Lamellocyte index, i.e., number of GFP-labelled cells relative to the size of the lymph gland (n=15). Statistical analysis by ANOVA followed by Tukey's multiple comparison test; ***p<0.001; significant differences are colour coded.

The online version of this article includes the following source data and figure supplement(s) for figure 9:

**Source data 1.** Raw data and statistical analysis.

**Figure supplement 1.** Pkc53E[Δ28] is sensitive to wasp infestation.

**Figure supplement 1—source data 1.** Raw data and statistical analysis.

**Figure supplement 2.** Representative images for the various settings.

*et al., 2019*). Our work now reveals that a parasitoid wasp attack causes a phosphorylation of Su(H) on S269 as means of an efficient and reversible way to inhibit Notch activity. This idea is consistent with earlier observations, whereby wasp parasitism quenched the expression of a Su(H)-lacZ reporter (*Small et al., 2014*). Moreover, we provide evidence that Pkc53E is involved in Su(H) phosphorylation during blood cell development as well as during parasitoid wasp defense in *Drosophila*. Moreover, the involvement of PKCs in wasp defense is not completely unexpected given their well-documented role

in the immune responses of mammals and *Drosophila* alike. For example, *Drosophila* flies ensure a healthy gut-microbiota homeostasis by modulating DUOX activity as a pathogen-specific defense line (reviewed in *Kim and Lee, 2014*). DUOX activation and ROS production induced by gut pathogens was shown to be mediated by Pkc53E and $Ca^{2+}$ via phospholipase Cβ (*Ha et al., 2009*; *Lee et al., 2015*). Sterile wounding of the *Drosophila* embryo triggers an instantaneous $Ca^{2+}$ flash followed by hydrogen peroxide production (*Razzell et al., 2013*). It is conceivable that the epidermal breach by the wasp's sting similarly induces a $Ca^{2+}$ flash that may spark Pkc53E activation as a result. However, PKCs may be activated directly by oxidative modification even in the absence of $Ca^{2+}$. Moreover, they may promote endogenous ROS production in a positive feedback loop (reviewed in *Cosentino-Gomes et al., 2012*), that could support the systemic response in the larval lymph gland.

The role of PKCs in the mammalian hematopoietic and immune systems is well documented. PKCs in this context act redundantly, and presumably, this holds also true for *Drosophila* where several PKCs exist. Upon activation, PKCs may translocate into the nuclear compartment, where they can directly influence gene expression programs, e.g., by piloting chromatin factors as substrates, pivotal for regulating immune cell differentiation (reviewed in *Lim et al., 2015*). Whereas *Drosophila* Pkc53E appears primarily cytoplasmic, its nuclear presence in some hemocytes is consistent with acting directly on Su(H) once activated (*Figure 8A*). Phosphorylation, however, may also occur at the membrane or in the cytoplasm, since Su(H) is imported into the nucleus together with its co-regulators (*Wolf et al., 2019*). PKCs, including PKCα, are present in CD34+ long-term hematopoietic stem cells. Moreover, specific roles in both the myeloid and the multilymphoid lineages are known. Notably, some PKC isoforms including PKCα may play a role in Notch-dependent T-cell development. Notch pathway activity is indispensable for T-cell lineage commitment of early thymic progenitors, however, is rapidly downregulated after the β-selection phase. As treatment with phorbol ester upregulates the transcription of TCRα and -β chains, a role of PKC in Notch-dependent commitment of αβ T-cells has been proposed (reviewed in *Altman and Kong, 2016*; *Yui and Rothenberg, 2014*). During initial T-cell development, Notch1 signalling increases in intensity, however, is abruptly downregulated at the transcriptional level following β-selection. This rapid downregulation is mediated by the inhibition of the E proteins (*Yashiro-Ohtani et al., 2009*), however, might in addition involve the phosphorylation of CSL by PKCα. Cell fate in more mature thymocytes is then fixed by the silencing of Notch target genes through chromatin modulators (reviewed in *Yui and Rothenberg, 2014*).

Appropriate silencing of Notch signalling activity in the course of T-cell development is of utmost importance, as prolonged Notch activity during β-selection predisposes the T-cells to leukemic transformation. Phosphorylation of CSL proteins by PKCα kinase offers a way for a timely and reversible deactivation of Notch signals not only in *Drosophila* but also in the mammalian system. In fact, the amino acid sequences harbouring the respective Serine residue in the CSL BTD are completely conserved between vertebrates and invertebrates (*Wilson and Kovall, 2006*; *Nagel et al., 2017*), raising the possibility of a likewise regulatory mechanism in mammalian hematopoiesis and immunity. Indeed, respective phosphorylation of the human CSL protein at the homologous position S195 was observed in human embryonic stem cells, where differentiation was induced by phorbol ester treatment (*Rigbolt et al., 2011*), consistent with an involvement of PKCs in this context as well. In conclusion, our work uncovers an important role for PKC-mediated downregulation of CSL activity, thereby ensuring blood cell homeostasis and an appropriate immune response to parasitoid wasp infestation in *Drosophila*. Future work may uncover, whether similar mechanisms apply to mammalian hematopoiesis and immunity as well.

## Materials and methods

Key resources table, see Appendix 1.

### Maintenance of parasitoid wasps and infection assay of *D. melanogaster*

*L. boulardi*, *L. heterotoma*, and *A. japonica* were kindly provided by B Häußling and J Stökl, Bayreuth, Germany (*Weiss et al., 2015*). Wasp species were co-cultured with wild-type *Drosophila* larvae at room temperature. To this end, about forty 3- to 5-day-old female wasps and twenty male wasps were co-incubated with second instar larvae for 5–7 days at room temperature. Every other day, fresh drops

of honey water were added to the vial plug for feeding the wasps. After two weeks, all hatched flies were discarded. Wasps emerge about 30 days after the infestation.

For the infection assays, 50–100 staged *Drosophila* late second/early third instar larvae of the respective genotype were transferred onto apple juice plates with fresh yeast paste. 30 female and 20 male wasps aged between 3 and 6 days were added to the larvae, allowing to infect them for 4–6 hr. Afterwards, wasps were removed and larvae were allowed to develop further in vials with normal fly food. Wasps were only used once for each infection. Only infected larvae containing wasp egg were used for the subsequent experiments. After infestation, hemolymph or lymph glands were prepared at the time points indicated for the particular application, or the survival rate of wasps versus *Drosophila* imago was recorded.

## Fly work and genetic analyses

Fly crosses were performed with 30–40 virgin females and 20 males to avoid overcrowding and stress. Combination/recombination of fly stocks was monitored by PCR genotyping using primers listed in *Supplementary file 5*. A complete list of the kinase mutant flies tested in the 'larval kinase screen' is found in *Supplementary file 3*. As reporter lines served *atilla*-GFP (BL23540) and *PPO3*-Gal4 UAS mCD8-GFP (named *PPO3::GFP*, *Dudzic et al., 2015*), *hmlΔ*-Gal4 UAS-GFP (named *hml::GFP* herein, BL30142) (*Sinenko and Mathey-Prevot, 2004*) and *He*-Gal4 UAS-GFP (BL8700) (*Zettervall et al., 2004*). For Gal4/UAS-based overexpression and RNAi-mediated knockdown, we used *hml*-Gal4 (BL30141), *lz*-Gal4 (*Lebestky et al., 2000*; obtained from M Crozatier, Université de Toulouse, France), UAS-*μMCas9* (VDRC 340002), UAS-HA-*Pkc53E* (this study), UAS-*white-RNAi* (BL31231), UAS-*Pkc53E*-RNAi[27491] (BL27491), UAS-*Pkc53E*-RNAi[34716] (BL34716), UAS-*sgRNA-Pkc53E* (VDRC341127), and UAS-*N-RNAi* (BL7078). The strain *vasa-φC31*, *96E-attB*/TM3 (*Bischof et al., 2007*) served for the generation of UAS-HA-Pkc53E flies. Su(H) controls and mutants comprised: *Su(H)*[gwt], *Su(H)*[gwt-mCh], *Su(H)*[S269A], and *Su(H)*[S269D]/*CyO-GFP* (*Praxenthaler et al., 2017*; *Frankenreiter et al., 2021*). *Su(H)*-[S269A-mCh] flies were produced in this study by a C-terminal in frame fusion of mCherry to the *Su(H)*[S269A] mutant gene followed by genomic integration of the construct via gene engineering as outlined before (*Praxenthaler et al., 2017*).

## Generation of the UAS-HA-Pkc53E fly line

*Pkc53E* cDNA (DGRC GH03188) was PCR-amplified and subcloned via *Xho*I/*Xba*I in a modified pBT-HA vector, harbouring three copies of an HA-Tag generated via annealed oligos cloned into *Acc*65I/*Xho*I to generate pBT-3xHA-*Pkc53E*. HA-*Pkc53E* was then shuttled via *Acc*65I/*Xba*I in likewise opened pUAST-attB vector (*Bischof et al., 2007*). All cloning steps were sequence verified. Primers used for cloning are included in *Supplementary file 5*. Transgenic fly lines were then generated with the help of the φC31 integrase-based system using 96E as landing site (*Bischof et al., 2007*).

## **Analyses of *Drosophila* hematopoetic cells and tissues**

### Recording total hemocyte numbers

Hemocytes were visualised by GFP fluorescence. To this end, *Su(H)*[gwt], *Su(H)*[S269A], and *Pkc53E*[Δ28] alleles, respectively, were combined with *hml*-Gal4 UAS-GFP (BL30142) and *He*-Gal4 UAS-GFP (BL8700) that together label the vast majority of hemocytes (*Petraki et al., 2015*). Larvae were vortexed at maximum speed with glass beads for 2 min, and the hemolymph was collected individually in 20 μl PBS. A fourth of the hemolymph was distributed in six Pap-pen wells of about 2 mm diameter. To ease staging and to avoid overcrowding, 4 hr egg collections were used, and larvae developed in batches of about 100 animals at 25°C until wandering third instar larval stage (ca. 120 hr after egg laying [AEL]). In case of wasp challenge, staged early L3 larvae were infested (ca. 90 hr AEL) and bled 20–32 hr thereafter to determine total hemocyte numbers. Only larvae containing wasp eggs were examined. GFP-positive hemocytes were visualised by epi-fluorescence microscopy on an Axioskop II (Zeiss, Jena), pictured with an EOS 700D camera (Canon, Japan). For counting cells, we followed earlier descriptions (*Petraki et al., 2015*). For reproducibility, the 'trainable Weka segmentation' plugin was used to demarcate cells from background (*Eibe et al., 2016*), followed by the 'analyse particles' plugin of ImageJ. Each Pap-pen well count was taken as a technical replicate for the individual larvae to determine the average hemocyte number per μl, expanded by 20 for the total hemocyte number per larva.

## Determination of sessile larval crystal cells

Larval crystal cells were counted according to *Frankenreiter et al., 2021*. Briefly, staged wandering third instar larvae of the respective genotype were heated to 60°C for 10–12 min. Pictures of the posterior dorsal side were taken with a Pixera camera (ES120, Optronics, Goleta, CA, USA) mounted to a stereo-microscope (Wild M3Z, Leica, Wetzlar, Germany) with Pixera Viewfinder 2.5. Melanised crystal cells appear as black dots, and were counted in the last two larval segments with *ImageJ* 1.51 software using *Cell Counter* tool. At least 20 larvae were scored for the statistical evaluation. For the PMA/STAU-feeding experiments, 20–30 developmentally synchronised second instar larvae were selected and grown for 24 hr in complete dark at 25°C on fly food with 200 μl of 1 mM PMA or PMA plus 0.2 mM STAU added to the surface of the fly food. As PMA and STAU were dissolved in DMSO before further dilution, controls were exposed accordingly to 10% DMSO on the fly food. Subsequently, wandering third instar larvae were heated and analysed as above.

## Visualisation and quantification of lamellocytes

The lamellocyte specific reporters *atilla*-GFP or *PPO3*::GFP strains were re/combined with *Su(H)^{gwt}*, *Su(H)^{S269A}*, or *Pkc53E^{Δ28}* alleles by genetic means. Late second/early third larval instars were infested by *L. boulardi* and the number of lamellocytes was determined 2 days later and compared with those observed in non-infested larvae. Larvae were washed thoroughly in cold PBS and dried with a tissue and teared apart. The hemolymph of 10 larvae each was collected with a 20 μl Microloader tip (Eppendorf, Hamburg, Germany) and placed on a slide with 7 μl of Vectashield mounting medium containing DAPI. GFP-positive cells, i.e., lamellocytes were counted in relation to the total number of DAPI-labelled hemocytes with a Zeiss Axioskop II and a PlanNeofluar ×20 objective. 8–10 independent bleedings were performed each.

## Pkc53E expression in hemocytes

*Pkc53E-eGFP* flies are derived from a protein trap and express endogenously a respective fusion protein (*Lee et al., 2018*). The hemolymph of three to five third instar *Pkc53E-eGFP* larvae at around 100 hr AEL, non-infested or infested at 80 hr AEL overnight with *L. boulardi*, was collected as described above in 20 μl cold PBS. Hemocytes were allowed to settle for 5–10 min in 500 μl cold PBS onto a round 18 mm glass slide placed in a 12-well microtiter plate. After fixation in 1 ml 4% paraformaldehyde in PBS for 15 min at room temperature, three washes with PBS plus 0.3% Triton X-100 (PBX), and a pre-incubation step in 500 μl 4% normal donkey serum in PBX for 45 min, cells were incubated overnight at 4°C with primary antibodies in 4% donkey serum in PBX (anti-GFP 1:100; anti-He 1:50; anti-Hnt 1:20, anti-mys 1:10, anti-PPO1 1:3; anti-Pzg 1:500), rhodamine-coupled phalloidin (1:200-1:400). Depending on the combination, staining with wheat germ agglutinin (WGA, 1:200) was performed for 15 min (hnt) and 60 min (PPO1) at room temperature. Three further washing steps and a pre-incubation step as above were followed by incubation for 2 hr at room temperature with suitable secondary antibodies (1:200) in the dark, followed by three additional washing steps. Cells were mounted in Vectashield by placing the round coverslip upside down on a glass slide, and pictures taken with a Bio-Rad MRC1024 coupled to Zeiss Axioskop with a PlanNeofluar ×63 objective using LaserSharp software 2000 (Zeiss, Jena, Germany).

## Immunostaining and documentation of larval lymph glands

Larval lymph glands were prepared 24–36 hr after wasp infection and treated as described before (*Frankenreiter et al., 2021*). For comparison, non-infested lymph glands were prepared. Primary antibodies used for staining: mouse anti-Hnt for crystal cells (1:20) and guinea pig anti-Pzg as nuclear marker (1:500). GFP signals were monitored directly. Secondary fluorescent antibodies were from Jackson ImmunoResearch Laboratories (1:250 each). Mounted tissue was documented with a Zeiss Axioskop coupled with a Bio-Rad MRC1024 confocal microscope using LaserSharp software 2000 (Zeiss, Jena, Germany). For statistical evaluation at least 12 primary lobes were documented and statistically analysed by using *Image J* software (*Schindelin et al., 2012*). Indices represent the number of cells in relation to the size/area of the tissue (in pixel) × 10,000.

## RNA expression analyses

### RT-PCR of Pkc53E$^{Δ28}$ null mutants and in hemocytes

Poly(A)$^+$ RNA was isolated from 50 third instar larvae (*Pkc53E$^{Δ28}$* and *y$^1$w$^{67c23}$*) using PolyATract System Kit 1000 according to the manufacturer's protocol, followed by a 10 min DNase I treatment at 37°C. Subsequent cDNA synthesis was conducted with qScriber cDNA Synthesis Kit according to the supplier's protocol. For amplification, a *Pkc53E* primer pair overlapping the last three introns was chosen (Pkc53E_RT-PCR UP and Pkc53E_RT-PCR LP). Tubulin 56D primers (Tub56D_229 UP and Tub56D_507 LP) served as internal controls. For primers, see *Supplementary file 5*. In order to monitor *Pkc53E* expression in hemocytes, hemolymph was derived from 20 third instar *Su(H)$^{gwt}$* larvae, and poly(A)$^+$ RNA isolated using the Dynabeads micro mRNA-Kit, with an on-beads DNase I digest, otherwise following the above protocol with primer pair Pkc53E_RT-PCR UP and Pkc53E_RT-PCR LP.

### Quantification of NRE-GFP transcription in hemocytes

*Su(H)$^{gwt}$* or *Su(H)$^{S269A}$* stocks were genetically combined with NRE-GFP (BL30728). Hemolymph was collected from 15 to 30 early third instar larvae of each genotype, infested with *L. boulardi* for 6 hr at 72 hr AEL as described above, to be compared with non-infested *Su(H)$^{gwt}$* control. Poly(A$^+$) RNA was isolated directly thereafter (0–6 hr value) or 24 hr later (24–30 hr value) with Dynabeads mRNA (micro) Kit from the cells lysed in 200 µl lysis and binding buffer according to the manufacturer's protocol, followed by an on-beads DNase I digest for 10 min at 37°C. After two washing steps, mRNA was eluted with 25 µl 10 mM Tris-HCl pH 8. cDNA synthesis was conducted with 15 µl using the qScriber cDNA Synthesis Kit according to the supplier's protocol. Real-time qPCR was performed as described before using the Blue S'Green qPCR Kit and the MIC magnetic induction cycler (bms, Australia), including target and no-template controls (*Kober et al., 2019*). Four biological replicates with two technical replicates were each conducted. Results were compared to the ubiquitously expressed genes *cyp33* and *Tbp* as reference. Primer pairs are listed in *Supplementary file 5*. The micPCR software version 2.12.7 was used for relative quantification of the data, based on REST and taking target efficiency into account (*Pfaffl et al., 2002*).

## Determination of kinases and kinase assays

### Screening of protein kinase candidates in silico and in vitro

To search for potential candidates in silico, GPS3.0 software was used at the lowest threshold levels, including the 40 kinases with the highest difference between score and cut-off value (*Xue et al., 2011*). The corresponding *Drosophila* kinases were determined with the help of flybase according to *Morrison et al., 2000*. For the in vitro screen, a 0.5 kb cDNA fragment (741–1242) encoding the Su(H) BTD (codons 247–414) was PCR-amplified and cloned via *Bam*HI/*Eco*RI into pGEX-2T vector (*Smith and Johnson, 1988*) for bacterial expression and purification of the BTD-GST fusion protein. Primers used for cloning are included in *Supplementary file 5*. ProQinase GmbH (Freiburg, Germany) provided the 'KinaseFinder assay service'. Briefly, BTD-GST and $^{33}$P-ATP served as substrates for 245 human Ser/Thr kinases in multi-well plates, analysed in a microplate scintillation reader. (A) Activity of each kinase was determined, (B) corrected for substrate background activity, and (C) auto-phosphorylation (kinase activity without substrate). A ratio value between phosphorylation of BTD-Su(H) and kinase auto-phosphorylation ≥1 (A-B/C) was considered as phosphorylating.

### In vitro ADP-Glo kinase assay

*Drosophila* pBT-3xHA-*Pkc53E* was mutated to generate the pseudo-activated form *Pkc53E$^{EDDD}$* (A34E/T508D/T650D/S669D) stepwise by site-directed mutagenesis using the Q5 Site directed Mutagenesis Kit. Primers used for mutagenesis are included in *Supplementary file 5*. *Pkc53E* as well as *PKC53E$^{EDDD}$* were then shuttled into a modified pMAL vector (*Riggs, 1994*), where additional restriction sites for *Acc*65I, *Sac*II, and *Xho*I had been included in the multiple cloning site via primer annealing. The MBP-Pkc53E and MBP-Pkc53E$^{EDDD}$ fusion proteins were bacterially expressed and purified with Amylose resin. Additionally, activated human kinase PKCα was obtained as a positive control. The PKCα pseudosubstrate PS (RFARLG**S**LRQKNV) (*Kochs et al., 1993*), the wild-type Su(H) peptide S$^{wt}$ (ALFNRLR-**S**QTVSTRY), and the phospho-deficient peptide S$^{SA}$ (ALFNRLR**A**QTVSTRY) were obtained (peptides & elephants, Hennigsdorf, Germany).

To test kinase activity, the ADP-Glo Kinase Assay system was used. Kinase assay reactions were performed in 96-well plates in a volume of 25 µl in the dark. Each reaction contained 100 µM of a kinase substrate peptide, 150 ng purified kinase, and 500 µM ultra-pure ATP. To stimulate kinase activity, 150 nM PMA was added to some reactions. The mixture was filled up with kinase reaction buffer (40 mM Tris-HCl, 20 mM MgCl$_2$, 0.1 mg/ml BSA, pH 7.4) and incubated at room temperature for 1 hr in the dark. 25 µl ADP-Glo Reagent were added and incubated for 40 min to remove residual ATP. 50 µl of Kinase Detection Reagent was applied to convert ADP to ATP. The luminescent signal was measured after 45 min using GloMax Discover Microplate Reader (Promega, Madison, WI, USA), kindly provided by the Department of Zoology (190z), University of Hohenheim.

## NanoLC-ESI-MS/MS analysis of Su(H) peptides

Nano-LC-ESI-MS/MS experiments were performed by the Mass Spectrometry Unit at the Core Facility Hohenheim (640) on an Ultimate 2000 RSLCnano system coupled to a Nanospray Flex Ion Source and a Q-Exactive HF-X mass spectrometer (Thermo Fisher Scientific, Waltham, MA, USA). Peptides were separated with LTQ-Orbitrap XL coupled to a nano-HPLC operated under the control of XCalibur 4.1.31.9 software (Thermo Fisher Scientific, Waltham, MA, USA). For all measurements using the Orbitrap detector, internal calibration was as described before (*Olsen et al., 2005*). MS/MS spectra were analysed using Proteome Discoverer 2.2 (Thermo Fisher Scientific, Waltham, MA, USA), verified by manual inspection of the MS/MS spectra (*Voolstra et al., 2010*).

## Generation of a α-pS269 antiserum

Rabbit polyclonal p-S269 Su(H) antiserum was generated by DAVIDS Biotechnology GmbH (Regensburg, Germany) using the synthetic phospho-peptide NRLR**pS**QTVSTRYLHVE. Phospho-specific antibodies were enriched in a depletion step by affinity purification against the non-phosphorylated peptide. We note a very low affinity of the purified antiserum for Su(H) protein. The antiserum detects purified Su(H)-GST fusion proteins (wild-type, S269A as well as S269D) with low affinity in western blots, but neither native Su(H) in tissue nor hemocytes nor in western blots, except if phosphorylated and enriched by Trap Technology.

## Immunoprecipitation of mCherry and Myc-tagged Su(H)

400 adult heads or 25 larvae of each genotype were homogenised on ice in 220 µl buffer 1 (150 mM NaCl, 1% Triton X-100, 50 mM Tris-HCl pH 7.5, 0.1% SDS, supplemented with protease inhibitor cocktail) and incubated for 15 min. After a short spin, 20 µl of the supernatant was set aside as input fraction ('protein extract'). The residual supernatant was diluted with 300 µl wash buffer I (see buffer I, but without Triton X-100). 15 µl of equilibrated magnetic RFP-Trap Magnetic Agarose beads were added, incubated for 1 hr at 8°C and washed three times with wash buffer I. mCherry-trapped proteins were resolved on SDS-PAGE; western blots were probed with rabbit anti-mCherry (1:1000) and rat anti-HA (1:500). Goat secondary antibodies coupled with alkaline phosphatase (1:1000) were used for detection.

## HeLa cell culture experiments and Luciferase assays

### Generation of HSV-TK 2xMyc-Su(H)-VP16

Two myc-tags were added to *Su(H)* cDNA in pBT (*Maier et al., 2011*) by insertion of the two annealed oligonucleotides Myc-Tag UP and Myc-tag LP into the *Eco*RI site (for primers, see **Supplementary file 5**). The construct was subsequently shuttled via *Eco*RI/*Xho*I into pCDNA3.1. A VP16 activator domain was then cloned in frame at the C-terminus of 2xMyc-*Su(H)* by replacing the 829 bp *Bsp*EI/*Apa*I fragment of pCDNA3 2xMyc-*Su(H)* with a respective 1060 bp fragment of the pUAST *Su(H)-VP16* construct (*Cooper et al., 2000*). Then the CMV Promoter of pCDNA3 2xMyc-Su(H)-VP16 was replaced by the HSV-TK Promoter of the pRL TK Vector. To this end, the 1023 bp *Bgl*II/*Nhe*I fragment from pRL TK was cloned into the *Bgl*II/*Spe*I opened pCDNA3 *2xMyc-Su(H)*$^{VP16}$ construct.

### Transfection of HeLa cells and reporter assay

*RBPj*$^{KO}$ HeLa cells are *RBPj*-deficient HeLa cells (ATCC: CLL-2; DSMZ: Acc57; obtained from DSMZ), generated by CRISPR-Cas9 in the laboratories of F Oswald (University of Ulm) and T Borggrefe

(University of Giessen), regularly monitored for mycoplasma contamination, confirmed by PCR and sequencing (*Wolf et al., 2019*). *RBPj^{KO}* HeLa cells (#4.42) were cultivated and transfected as described (*Wolf et al., 2019*). The following constructs were used: pGL3 NRE-reporter (*Bray et al., 2005*), pCDNA3 HSV-TK 2xMyc Su(H)^{VP16}, and pRL TK. For the Luciferase assay, $1 \times 10^5$ HeLa *RBPj^{KO}* cells were seeded in each well of a 12-well cell culture plate. After 24 hr the cells were transfected with 500 ng pGL3 NRE-reporter, 460 ng pCDNA3 HSV-TK 2xMyc Su(H)^{VP16}, and 40 ng pRL-TK. 4 hr after the transfection the cells were treated with 162 nM PMA, 162 nM PMA plus 21.4 nM STAU or 21.4 nM STAU alone. Control cells were treated with the same volume of DMSO present in the other treatments. 14 hr later, cells were washed twice in PBS pH 7.4 and lysed in 75 µl 1x Passive Lysis Buffer. The Dual-Luciferase Reporter Assay was performed according to the manufacturer's instructions.

Su(H)-VP16 expression levels were detected with anti-Myc mAB (1:500) and anti-beta-tubulin as loading control (1:500). To this end, $5 \times 10^5$ HeLa *RBPj^{KO}* cells were transfected and treated as above, harvested and lysed 14 hr later in 300 µl binding buffer (20 mM HEPES pH 7.6, 150 mM $MgCl_2$, 10% glycerol, 0.05% NP-40, 1 mM DTT, ROCHE cOmplete ULTRA-tablet Mini protease inhibitor); 20 µl of each lysate were loaded for western blotting.

## Statistical analysis and documentation of data

Normality of the data was checked by a Shapiro-Wilk test using GraphPad Prism 9.0. In case of normally distributed data, a two-tailed analysis of variance (ANOVA) approach for multiple comparisons according to Tukey-Kramer's Honestly Significance Difference or an unpaired t-test was applied, and in the other cases the non-parametric Kruskal-Wallis sum test or Dunn's test for multiple comparisons. Quantification of RT-PCR data is based on REST, and was performed with the micPCR software version 2.12.7, taking target efficiency into account (*Pfaffl et al., 2002*). In the figures, p-values are presented as ***, $p<0.001$; **, $p<0.01$; *, $p<0.05$; not significant, $p \geq 0.5$; the exact p-values are given in the respective source data. Pictures were assembled using ImageJ, PhotoPaint, CorelDraw, and BoxPlotR software. In the box plots made by BoxPlotR, centre lines show the medians; box limits indicate the 25th and 75th percentiles as determined by R software; whiskers extend 1.5 times the interquartile range from the 25th and 75th percentiles, outliers are represented by dots (*Spitzer et al., 2014*). Number of sample points is given in the figure or legend.

## Acknowledgements

We are deeply grateful to Benedikt Häußling and Johannes Stökl (University of Bayreuth, Germany) for sending us all wasp species used in this study and for giving SD and LF a basic course in handling of the wasps. We thank Michèle Crozatier (Toulouse, France), David Hipfner (Montréal, Canada), Bruno Lemaitre (Lausanne, Switzerland), Sarah Bray (Cambridge, UK), Dieter Maier (Hohenheim, Germany), the Bloomington *Drosophila* Stock Center (BDSC, NIH P40OD018537) and the Vienna Drosophila Stock Center (VDRC) for numerous fly stocks. We acknowledge the Drosophila Genomics Resource Center (DGRC, NIH 2P40OD010949) for sending the Pkc53E cDNA, and the Developmental Studies Hybridoma Bank (DSHB), created by the NICHD of the NIH and maintained at the University of Iowa, Department of Biology, Iowa City, IA 52242 for providing several monoclonal antibodies. We thank Istvan Andó (Szeged, Hungary) for anti-Hemese antiserum, and Tina E Trenczek (Gießen, Germany) for anti-PPO1 antibodies. We very much acknowledge Franz Oswald (Ulm, Germany) and Tilman Borggrefe (Gießen, Germany) for the *RBPj^{KO}* HeLa cell line. We are indebted to Lisa Lermer for her help in tissue preparations and screening of larvae and Janika Scharpf for the purification of the Pkc53E proteins. We are grateful to Armin Huber, Department of Biochemistry, for the use of the Apotome microscope, to Axel Schweickert, Department of Zoology, for use of the GloMax Discover Microplate Reader and to Jens Pfannstiel at the Mass Spectrometry Unit (Core Facility of the University of Hohenheim) for the MS/MS and ESI spectra. We thank Dieter Maier for helpful comments on the manuscript. Funding. This work was supported by a grant of the German Science Foundation DFG to ACN (NA 427/5-1) and by the University of Hohenheim. The funders had no role in study design, data collection, and interpretation, or the decision to submit the work for publication.

# Additional information

## Funding

| Funder | Grant reference number | Author |
|---|---|---|
| Deutsche Forschungsgemeinschaft | NA 427/5-1 | Anja C Nagel |
| Universität Hohenheim | | Anja C Nagel |

The funders had no role in study design, data collection and interpretation, or the decision to submit the work for publication.

## Author contributions
Sebastian Deichsel, Resources, Formal analysis, Validation, Investigation, Visualization, Methodology, Writing – review and editing; Lisa Frankenreiter, Johannes Fechner, Resources, Formal analysis, Validation, Investigation, Methodology, Writing – review and editing; Bernd M Gahr, Resources, Investigation, Visualization, Methodology, Writing – review and editing; Mirjam Zimmermann, Resources, Formal analysis, Investigation, Writing – review and editing; Helena Mastel, Irina Preis, Investigation, Writing – review and editing; Anette Preiss, Formal analysis, Visualization, Writing – original draft, Writing – review and editing; Anja C Nagel, Conceptualization, Formal analysis, Supervision, Funding acquisition, Validation, Visualization, Methodology, Writing – original draft, Project administration, Writing – review and editing

## Author ORCIDs
Bernd M Gahr ⓘD https://orcid.org/0000-0002-5755-6603
Anette Preiss ⓘD http://orcid.org/0000-0002-6410-1586
Anja C Nagel ⓘD https://orcid.org/0000-0002-2733-3249

Reviewer #1 (Public review): https://doi.org/10.7554/eLife.89582.3.sa1
Reviewer #2 (Public review): https://doi.org/10.7554/eLife.89582.3.sa2
Reviewer #3 (Public review): https://doi.org/10.7554/eLife.89582.3.sa3
Author response https://doi.org/10.7554/eLife.89582.3.sa4

# Additional files

## Supplementary files
• Supplementary file 1. List of kinases predicted to recognise S269 in Su(H) as substrate in silico. Computationally determined candidate Ser/Thr kinases predicted to pilot Serine 269 in Su(H). The list contains the human and the corresponding *Drosophila* candidates.

• Supplementary file 2. List of kinases accepting the beta-trefoil domain (BTD) domain of Su(H) as substrate in vitro. List of Ser/Thr kinases that tested positive in accepting *Drosophila* Su(H) BTD as a substrate for phosphorylation in an in vitro assay. The list contains the 62 human and the corresponding 40 *Drosophila* candidates, highlighting the 10 from *Drosophila* also identified in the in silico screen.

• Supplementary file 3. Fly strains used for the larval crystal cell screen. This file contains a list of the mutant alleles and RNAi strains of the *Drosophila* Ser/Thr kinases screened for alterations in crystal cell numbers with identifier, reference and/or source (BL, Bloomington Drosophila Stock Center; VDRC, Vienna Drosophila Resource Center).

• Supplementary file 4. Larval crystal cell screen. This file contains the results from the larval crystal cell screen. The list displays the Ser/Thr kinases and the relevant controls tested in the screen, the alleles or RNAi settings used, the average crystal cell number of the tested mutant and the percentage gain or loss of crystal cells relative to the control, SD, and sample size. Heated larvae of kinase mutants and/or *hml*-Gal4::UAS-kinase-RNAi/UAS-kinase$^{DN}$ genotypes were counted for the appearance of melanised crystal cells (cc) in the last two segments. If UAS-transgenes were used, the number of crystal cells was compared between uninduced (UAS-line alone) and induced with *hml*-Gal4. Mutant genotypes were related to the *Su(H)$^{gwt}$* wild-type control.

• Supplementary file 5. List of oligonucleotides. This file contains a list of oligonucleotides used for cloning, mutagenesis, and verification of constructs, as well as for RT-PCR and qRT-PCR analyses, including PCR conditions.

• MDAR checklist

## Data availability

All data generated or analysed during this study are included in the manuscript and supporting files; source data files have been provided for all figures.

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

# Appendix 1

## Appendix 1—key resources table

| Reagent type (species) or resource | Designation | Source or reference | Identifiers | Additional information |
|---|---|---|---|---|
| Gene (*D. melanogaster*) | *Pkc53E* cDNA | *Drosophila* Genome Resource Center | GH03188 | |
| Recombinant DNA reagent | pBT-3xHA-Pkc53E | This paper | N/A, available upon request | HA-tagged Pkc53E subclone in pBT |
| Recombinant DNA reagent | *Pkc53E*$^{EDDD}$ | This paper | A34E/T508D/T650D/S669D | In vitro mutagenised 3xHA-Pkc53E cDNA |
| Gene (*D. melanogaster*) | Su(H) cDNA | **Maier et al., 2011** | N/A | |
| Recombinant DNA reagent | Su(H) BTD in pGEX-2T | This paper | N/A, available upon request | For bacterial expression of a BTD-GST fusion protein |
| Recombinant DNA reagent | 2xMyc-Su(H) in pCDNA3.1 | This paper | N/A, available upon request | myc-tagged version of Su(H) |
| Recombinant DNA reagent | HSV-TK 2xMyc-Su(H)-VP16 | This paper | N/A, available upon request | myc-tagged version of Su(H) with VP16 activation domain under HSV control |
| Transfected construct (*D. melanogaster*) | HSV-TK 2xMyc-Su(H)-VP16 in *RBPj*$^{KO}$ HeLa cells | **Wolf et al., 2019**; this paper | N/A | Assay on the influence of PMA/Stau on Su(H)-VP16 activity |
| Recombinant DNA reagent | pGL3 NRE | **Bray et al., 2005** | N/A | |
| Recombinant DNA reagent | pUAST Su(H)-VP16 | **Cooper et al., 2000** | N/A | |
| Recombinant DNA reagent | pUAST-attB | **Bischof et al., 2007** | RRID:DGRC_1419 | |
| Recombinant DNA reagent | pBT-3xHA | This paper | N/A, available upon request | Three HA-tags cloned into *Acc*65I/*Xba*I sites of pBT |
| Recombinant DNA reagent | pGEX-2T | **Smith and Johnson, 1988** | N/A | |
| Recombinant DNA reagent | pMAL | **Riggs, 1994** | N/A | |
| Recombinant DNA reagent | pCDNA3.1 | Invitrogen | Cat# V79020 | |
| Recombinant DNA reagent | pRL TK | Promega | Cat# E2241 | |
| Strain, strain background (*Leptopilina boulardi*) | *Leptopilina boulardi* | Häußling, J Stökl, Bayreuth | N/A | Parasitoid wasp |
| Strain, strain background (*Leptopilina heterotoma*) | *Leptopilina heterotoma* | Häußling, J Stökl, Bayreuth | N/A | Parasitoid wasp |
| Strain, strain background (*Asobara japonica*) | *Asobara japonica* | Häußling, J Stökl, Bayreuth | N/A | Parasitoid wasp |
| Genetic reagent (*D. melanogaster*) | *atilla*-GFP, i.e. w$^{1118}$; Mi{ET1}atilla$^{MB03539}$ | Bloomington *Drosophila* Stock Center | BDSC:23540 | |
| Genetic reagent (*D. melanogaster*) | *He*-Gal4 UAS-GFP, i.e. w*; P{He-GAL4.Z}85, P{UAS-GFP.nls}8 | Bloomington *Drosophila* Stock Center | BDSC:8700 | |
| Genetic reagent (*D. melanogaster*) | *hml*-Gal4, i.e. w$^{1118}$; P{Hml-GAL4.Δ}3/MKRS | Bloomington *Drosophila* Stock Center | BDSC:30141 | |
| Genetic reagent (*D. melanogaster*) | *hmlΔ*-Gal4 UAS-GFP, i.e. w$^{1118}$; P{Hml-GAL4.Δ}3, P{UAS-2xEGFP}AH3/MKRS | Bloomington *Drosophila* Stock Center | BDSC:30142 | |
| Genetic reagent (*D. melanogaster*) | *PPO3*-Gal4 UAS mCD8-GFP | **Dudzic et al., 2015** | N/A | |
| Genetic reagent (*D. melanogaster*) | *lz*-Gal4 | **Lebestky et al., 2000** | N/A | |
| Genetic reagent (*D. melanogaster*) | *NRE-GFP*, i.e. w$^{1118}$; P{NRE-EGFP.S}1 | Bloomington *Drosophila* Stock Center | BDSC:30728 | |
| Genetic reagent (*D. melanogaster*) | *Pkc53E-EGFP*, i.e. Pkc53E$^{MI05296-GFSTF.0}$ | Bloomington *Drosophila* Stock Center | BDSC:59413 | |
| Genetic reagent (*D. melanogaster*) | UAS-*white-RNAi*, i.e. y$^1$ v$^1$; P{TRiP.JF01574}attP2/TM3, Ser$^1$ | Bloomington *Drosophila* Stock Center | BDSC:31231 | |
| Genetic reagent (*D. melanogaster*) | UAS-*N-RNAi*, i.e. P{UAS-N.RNAi.P}14E, w* | Bloomington *Drosophila* Stock Center | BDSC:7078 | |
| Genetic reagent (*D. melanogaster*) | UAS-*sgRNA-Pkc53E* | Vienna *Drosophila* Resource Center | VDRC341127 | |

*Appendix 1 Continued on next page*

*Appendix 1 Continued*

| Reagent type (species) or resource | Designation | Source or reference | Identifiers | Additional information |
|---|---|---|---|---|
| Genetic reagent (*D. melanogaster*) | UAS-*µMCas9* | Vienna *Drosophila* Resource Center | VDRC 340002 | |
| Genetic reagent (*D. melanogaster*) | UAS-HA-*Pkc53E* | This study | N/A | HA-Pkc53E under UAS-control integrated at 96E (3R) |
| Genetic reagent (*D. melanogaster*) | *vasa-φC31; 96E-attB*/TM3 | **Bischof et al., 2007** | N/A | |
| Genetic reagent (*D. melanogaster*) | *Su(H)$^{gwt}$* | **Praxenthaler et al., 2017** | N/A | |
| Genetic reagent (*D. melanogaster*) | *Su(H)$^{gwt-mCh}$, i.e. y$^1$ w$^*$;* TI{TI}Su(H)$^{gwt-mCh}$ | **Praxenthaler et al., 2017** | BDSC:94607 | |
| Genetic reagent (*D. melanogaster*) | *Su(H)$^{S269A}$, i.e. y$^1$ w$^*$;* TI{TI}Su(H)$^{S269A}$ | **Frankenreiter et al., 2021** | BDSC:94609 | |
| Genetic reagent (*D. melanogaster*) | *Su(H)$^{S269D}$/CyO-GFP, i.e. y$^1$ w$^*$;* TI{TI}Su(H)$^{S269D}$/ CyO, P{GAL4-Hsp70.PB}TR1, P{UAS-GFP.Y}TR1 | **Frankenreiter et al., 2021** | BDSC:94610 | |
| Genetic reagent (*D. melanogaster*) | *Su(H)$^{S269A-mCh}$, i.e. y$^1$ w$^*$;* TI{TI}Su(H)$^{S269A-mCh}$ | This study | N/A | Knock-in allele of mCherry-tagged *Su(H)$^{S269A}$* into the native *Su(H)* locus |
| Genetic reagent (*D. melanogaster*) | Kinase mutant flies tested in the larval kinase screen are listed in **Supplementary file 3** | BDSC, VDRC, and various donors as indicated in **Supplementary file 3** | | |
| Cell line (*H. sapiens*) | *RBPj$^{KO}$* HeLa cells (origin is ATCC: CLL-2; DSMZ: ACC57) | **Wolf et al., 2019**; gift of F Oswald (University of Ulm) and T Borggrefe (University of Giessen) | N/A | Homozygous knockout of the RBPj gene |
| Antibody | Mouse monoclonal anti-Hnt, 1G9 | Developmental Studies Hybridoma Bank, developed by H Lipshitz | RRID: AB_528278 | IF(1:20) |
| Antibody | Mouse monoclonal anti-mys, CF.6G11 | Developmental Studies Hybridoma Bank, developed by D Brower | RRID: AB_528310 | IF(1:10) |
| Antibody | Mouse monoclonal anti-beta tubulin, E7 | Developmental Studies Hybridoma Bank, developed by M Klymkowsky | RRID: AB_2315513 | WB(1:500) |
| Antibody | Mouse monoclonal anti-myc 9B11 | Cell Signaling Techn. | RRID: AB_331783; Cat# 2276 | WB(1:500) |
| Antibody | Mouse monoclonal anti-Hemese | **Kurucz et al., 2003**; gift from I Andó, Szeged, Hungary | N/A | IF(1:50) |
| Antibody | Guinea pig polyclonal anti-Pzg | **Kugler and Nagel, 2007** | N/A | IF(1:500) |
| Antibody | Mouse monoclonal anti-PPO1, 12F6 | **Trenczek and Bennich, 1992**; gift from TE Trenczek, Giessen, Germany | N/A | IF(1:3) |
| Antibody | Mouse monoclonal anti-GST (8-326) | Invitrogen | S RRID: AB_10979611, Cat# MA4-004 | WB(1:1000) |
| Antibody | Rabbit polyclonal anti-pS269 | This paper, DAVIDS Biotechnology GmbH | N/A | WB(1:100) |
| Antibody | Rabbit polyclonal anti-GFP | Santa Cruz | RRID: AB_641123; Cat# sc-8334 | IF(1:100) |
| Antibody | Rabbit polyclonal anti-mCherry | GeneTex | RRID: AB_2721247; Cat# GTX128508 | WB(1:1000) |
| Antibody | Rat monoclonal anti-HA 3F10 | ROCHE | RRID:AB_390918 Cat# 11867423001 | WB(1:500) |
| Antibody | Donkey polyclonal anti-mouse IgG, Cy3 | Jackson ImmunoResearch Laboratories | RRID: AB_2315777 Cat# 715-165-151 | IF(1:200) |
| Antibody | Donkey polyclonal anti-mouse IgG, Cy5 | Jackson ImmunoResearch Laboratories | RRID: AB_2340820 Cat# 715-175-151 | IF(1:200) |
| Antibody | Donkey polyclonal anti- guinea pig IgG, Cy5 | Jackson ImmunoResearch Laboratories | RRID: AB_2340462 Cat# 706-175-148 | IF(1:200) |
| Antibody | Donkey polyclonal anti- rabbit IgG, FITC | Jackson ImmunoResearch Laboratories | RRID: AB_2315776 Cat# 711-095-152 | IF(1:200) |

*Appendix 1 Continued on next page*

*Appendix 1 Continued*

| Reagent type (species) or resource | Designation | Source or reference | Identifiers | Additional information |
|---|---|---|---|---|
| Antibody | Goat polyclonal anti-guinea pig IgG, Alexa Fluor 647 | Jackson ImmunoResearch Laboratories | RRID: AB_2337446; Cat# 106-605-003 | IF(1:200) |
| Antibody | Goat polyclonal anti-mouse IgG, FITC | Jackson ImmunoResearch Laboratories | RRID: AB_2338601; Cat# 1115-095-166 | IF(1:200) |
| Antibody | Goat polyclonal anti-rabbit IgG, alkaline phosphatase | Jackson ImmunoResearch Laboratories | RRID: AB_2337947; Cat# 111-055-003 | WB(1:1000) |
| Antibody | Goat polyclonal anti-rat IgG, alkaline phosphatase | Jackson ImmunoResearch Laboratories | RRID: AB_2338148; Cat# 112-055-003 | WB(1:1000) |
| Antibody | Goat polyclonal anti-mouse IgG, alkaline phosphatase | Jackson ImmunoResearch Laboratories | RRID: AB_2338528; Cat# 115-055-003 | WB(1:1000) |
| Other | Normal donkey serum | Jackson ImmunoResearch Laboratories | RRID: AB_2337258 Cat# 017-000-121 | IF(1:400) |
| Other | Normal goat serum | Jackson ImmunoResearch Laboratories | RRID: AB_2336990 Cat# 005-000-121 | IF(1:400) |
| Peptide, recombinant protein | Activated PKCα | ProQinase | Cat# 0222-0000-1 | |
| Peptide, recombinant protein | Activated Akt1 | ProQinase | Cat# 1379-0000-2 | |
| Peptide, recombinant protein | Activated GSK3 beta | ProQinase | Cat# 0310-0000-1 | |
| Peptide, recombinant protein | Activated S6K | ProQinase | Cat# 0318-0000-2 | |
| Peptide, recombinant protein | CAMK2D | Invitrogen | NP_742113 | |
| Peptide, recombinant protein | Pseudosubstrate | peptides & elephants | PS | RFARLGSLRQKNV |
| Peptide, recombinant protein | Su(H) peptide | peptides & elephants | $S^{wt}$ | ALFNRLRSQTVSTRY |
| Peptide, recombinant protein | Su(H)SA peptide | peptides & elephants | $S^{SA}$ | ALFNRLRAQTVSTRY |
| Peptide, recombinant protein | Su(H) phosphopeptide | DAVIDS Biotechnology GmbH | N/A | NLRLpSQTVSTRYLHVE |
| Commercial assay or kit | RFP-Trap Magnetic Agarose | ChromoTek | Cat# rtma-20 | |
| Commercial assay or kit | Amylose resin | New England Biolabs GmbH | Cat# E8021S | |
| Commercial assay or kit | ADP-Glo Kinase Assay | Promega | Cat# V6930 | |
| Commercial assay or kit | PolyATract System Kit 1000 | Promega | Cat# Z5400 | |
| Commercial assay or kit | Dynabeads mRNA DIRECT micro purification kit | Invitrogen, Thermo Fisher | Cat# 61021 | |
| Commercial assay or kit | qScriber cDNA Synthesis Kit | highQu | Cat# RTK0104 | |
| Commercial assay or kit | Q5 Site directed Mutagenesis Kit | New England Biolabs GmbH | Cat# E0554S | |
| Commercial assay or kit | Blue S'Green qPCR Kit | Biozym | Cat# 331416 | |
| Commercial assay or kit | Dual-Luciferase Reporter Assay | Promega | Cat# E1910 | |
| Commercial assay or kit | Pap-pen | Kisker Biotech | Cat# MKP-1 | |
| Chemical compound, drug | DAPI | Cell Signaling Techn. | Cat# 4083 | (1 µg/ml) |
| Chemical compound, drug | DNase I | New England Biolabs GmbH | Cat# M0303 | (2 U/µl) |
| Chemical compound, drug | PMA, Phorbol-12-myristat-13-acetat | Sigma-Aldrich | Cat# P8139-1MG | (1 mM) |
| Chemical compound, drug | Staurosporine | Sigma-Aldrich | Cat# S4400-1MG | (0.2 mM) |
| Chemical compound, drug | Protease inhibitors, cOmplete ULTRA-tablets Mini | Roche | Cat# 5892791001 | (1 tablet/10 ml) |
| Chemical compound, drug | PhosSTOP (Phosphatase inhibitor) | Roche | Cat# 4906837001 | (1 tablet/10 ml) |
| Chemical compound, drug | Wheat germ agglutinin (WGA), Alexa Fluor 647 conjugate | Fisher Scientific | Cat# 11510826 | (1:200) |

*Appendix 1 Continued on next page*

*Appendix 1 Continued*

| Reagent type (species) or resource | Designation | Source or reference | Identifiers | Additional information |
|---|---|---|---|---|
| Chemical compound, drug | Phalloidin, coupled to rhodamine | Invitrogen, Thermo Fisher | Cat# R415 | (1:200-1:400) |
| Chemical compound, drug | Vectashield | Biozol | Cat# VEC-H_1000 | Mounting medium |
| Software, algorithm | GPS3.0 software | *Xue et al., 2011* | | |
| Software, algorithm | *ImageJ* 1.51 | *Schindelin et al., 2012* | https://imagej.nih.gov/ij/ | |
| Software, algorithm | GraphPad Prism version 9.0 | GraphPad Software, Inc | https://www.graphpad.com/ | |
| Software, algorithm | MIC PCR software version v2.12.7 | bms/Biozym | Cat# 68MiC-HRM | |
| Software, algorithm | Weka machine learning and data analysis software version 3.8 | *Eibe et al., 2016* | https://waikato.github.io/weka-site/index.html | |
| Sequence-based reagent | *Supplementary file 5*, oligonucleotides | Microsynth AG | | |

