## [Editor Report · eLife Assessment]

This **valuable** study focuses on the regulation of Notch signaling during the immune response in Drosophila. The authors provide **solid** evidence in support of roles for Su(H) and Pkc53E-induced phosphorylation in *Drosophila* immunity. The work will be of interest to colleagues in immunity and receptor signaling.

---

## [Referee Report · Reviewer #1 (Public review)]

The authors previously showed in cell culture that Su(H), the transcription factor mediating Notch pathway activity in *Drosophila*, was phosphorylated on S269 and they found that a phospho-deficient Su(H) allele behaves as a moderate gain of Notch activity in flies, notably during blood cell development. Since downregulation of Notch signaling is important for the production of specialized blood cell types (lamellocytes) in response to wasp parasitism, the authors hypothesized that Su(H) phosphorylation might be involved in this cellular immune response.

Consistent with their hypothesis, the authors now show that Su(H)S269A knock-in flies display a reduced response to wasp parasitism and that Su(H) is phosphorylated upon infestation. Using in vitro kinase assays and a genetic screen, they identify the PKCa family member Pkc53E as the putative kinase involved in Su(H) phosphorylation and they show that Pkc53E can bind Su(H). They further show that Pkc53E deficit or its knock-down in larval blood cells results in similar blood cell phenotypes as Su(H)S269A and their epistatic analyses indicate that Pkc53E acts upstream of Su(H). Finally, they show that Pkc53E mutants aslo display a compromised immune response to wasp parasitism.

Strengths

The manuscript is well presented and the experiments are sound, with a good combination of genetic and biochemical approaches and several clear phenotypes backing the main conclusions. Notably Su(H)S269A mutation strongly reduces lamellocyte production. Moreover, the epistatic data are convincing, notably concerning the relationship between Notch/Su(H) and Pkc53E for crystal cell production.

Even though it is not fully established, the overall model is credible and interesting. In addition, it opens further avenues of research to study the activation of Pkc in response to an immune challenge.

Weaknesses

Apparently, the hypothesis that Pkc53E is required for Su(H) phosphorylation in vivo could not be directly tested due to the lack of an appropriate tool (the specificity and sensitivity of the current anti-pS269 antibody was insufficient).

Also, the poor immune response of Pkc53E mutant might rather be linked to their constitutively reduced circulating blood cell number than to a deficit in Notch/Su(H) down-regulation following wasp infestation.

---

## [Referee Report · Reviewer #2 (Public review)]

The current draft by Deischel et.al., describes the role of Pkc53E in the phosphorylation of Su(H) to down regulate its transcriptional activity to mount a successful immune response upon parasitic wasp-infection. Overall, I find the study interesting and relevant especially the identification of Pkc53E in phosphorylation of Su(H) is very nice. The authors have proved the central idea linking phosphorylation of Su(H) via Pkc53E to implying its modulation of Notch activity to mount a robust immune response is now well addressed in its entirety and I find the paper indeed very interesting.

Comments on revised version:

The authors have addressed all pending concerns and I have no further comments. I indeed complement the authors for their wonderful piece of work.

---

## [Referee Report · Reviewer #3 (Public review)]

Diechsel et al. provide important and valuable insights into how Notch signaling is shut down in response to parasitic wasp infestation in order to suppress crystal cell fate and favor lamellocyte production. The study shows that CSL transcription factor Su(H) is phosphorylated at S269A in response to parasitic wasp infestation and this inhibitory phosphorylation is critical for shutting down Notch. The authors go on to perform a screen for kinases responsible for this phosphorylation and have identified Pkc53E as the specific kinase acting on Su(H) at S269A. Using analysis of mutants, RNAi and biochemistry-based approaches the authors convincingly show how Pkc53E-Su(H) interaction is critical for remodeling hematopoiesis upon wasp challenge. I find the study interesting, and the data presented supports the overall conclusions made by the authors. The authors have addressed all my comments satisfactorily in the revised submission.

Strengths:

The manuscript is well presented, and the conclusions made are backed by genetic, biochemical and molecular biology-based approaches. Overall, the authors convincingly demonstrate how Pkc53E mediated phosphorylated of Su(H) shuts down Notch signaling during wasp infestation in *Drosophila*.

Weaknesses:

The exact molecular trigger for activation of Pkc53E is still uncharacterized and it would be interesting to know how Pkc53E gets activated during wasp infestation and whether Pkc53E gets activated turning down Notch in other stress induced scenarios.

The authors have addressed comments satisfactorily. Overall, I think the findings are interesting and would be useful to the field of developmental biology and immunology and address an important gap in the field. The most significant conclusion from the work is how Notch acts as a molecular switch during parasitic wasp infestation.

---

## [Author Response]

The following is the authors’ response to the original reviews.

**Reviewer #1 (Public Review):**
The authors previously showed in cell culture that Su(H), the transcription factor mediating Notch pathway activity, was phosphorylated on S269 and they found that a phospho-deficient Su(H) allele behaves as a moderate gain of Notch activity in flies, notably during blood cell development. Since a downregulation of Notch signaling was proposed to be important for the production of a specialized blood cell types (lamellocytes) in response to wasp parasitism, the authors hypothesized that Su(H) phosphorylation might be involved in this cellular immune response.Consistent with their hypothesis, the authors show that Su(H)S269A knock-in flies display a reduced response to wasp parasitism and that Su(H) is phosphorylated upon infestation. Using in vitro kinase assays and a genetic screen, they identify the PKCa family member Pkc53E as the putative kinase involved in Su(H) phosphorylation and they show that Pkc53E can bind Su(H). They further show that Pkc53E deficit or its knock-down in larval blood cells results in similar blood cell phenotypes as Su(H)S269A, including a reduced response to wasp parasitism, and their epistatic analyses indicate that Pkc53E acts upstream of Su(H).StrengthsThe manuscript is well presented and the experiments are sound, with a good combination of genetic and biochemical approaches and several clear phenotypes which back the main conclusions. Notably Su(H)S269A mutation or Pkc53E deficiency strongly reduces lamellocyte production and the epistatic data are convincing.WeaknessesThe phenotypic analysis of larval blood cells remains rather superficial. Looking at melanized cells is a crude surrogate to quantify crystal cell numbers as it is biased toward sessile cells (with specific location) and does not bring information concerning the percentage of blood cells differentiated along this lineage.In Su(H)S269A knock-in or Pkc53E zygotic mutants, the increase in crystal cells in uninfected conditions and the decreased capacity to induce lamellocytes following infection could have many origins which are not investigated. For instance, premature blood cell differentiation could promote crystal cell differentiation and reduce the pool of lamellocytes progenitors. These mutations could also affect the development and function of the posterior signaling center in the lymph gland, which plays a key role in lamellocyte induction.Similarly, the mild decrease on resistance to wasp infestation (Fig. 2A) could reflect a constitutive reduction in blood cell numbers in Su(H)S269A larvae rather than a defective down-regulation of Notch activity.

We fully agree with the reviewer that sessile crystal cells counts are a coarse approach to capture hemocytes. However, they allowed the screening of numerous genotypes in the course of our kinase candidate screen. We recorded the hemocyte numbers in the various genetic backgrounds and with regard to wasp infestation. There was no significant difference between Su(H)S269A and Su(H)gwt control, independent of infection. This is in agreement with earlier observations of unchanged plasmatocyte numbers in *N* or *Su(H)* mutants compared to the wild type (Duvic et al., 2002). We noted, however, a small drop in hemocyte numbers in Su(H)S269D and a strong one in Pkc53ED28 mutants in both conditions relative to control. Presumably, Pkc53E has a more general role in blood cell development, which we have not further analysed. The results were included in new Figure 1_S1 and Figure 9_S1 supplements. Based on the link between hemocyte numbers and wasp resistance (e.g. [56]), we cannot exclude that the lowered resistance of Pkc53ED28 mutants regarding wasp attacks is partly due to reduced hemocyte numbers, albeit we did not see significant differences between either Su(H)S269A, nor Pkc53ED28 nor the double mutant. We have included this notion in the text.

Lamellocytes arise in response to external challenges like parasitoid wasp infestation by trans-differentiation from larval plasmatocytes, and by maturation of lamellocyte precursors in the lymph gland, yet barely in the Su(H)S269A and Pkc53ED28 mutants.

We find it hard to envisage, however, that a premature differentiation of plasmatocytes into crystal cells in our case could deplete the pool of lamellocyte progenitors in the hemolymph. (Is there a precedent?). Crystal cells make up about 5% of the hemocyte pool; they are increased max. 2 fold in the Su(H)S269A and Pkc53E mutants. Even if these extra crystal cells (now ~10%) had arisen by premature differentiation, there should be still enough plasmatocytes (~ 80%) remaining with a potential to further divide and transdifferentiate into lamellocytes.

Indeed, we cannot exclude an effect of the Su(H)S269A mutant on the development and function of the posterior signaling center of the lymph gland. We noted, however, a slight but significant enlargement of the PS in the Su(H)S269A mutant, that to our understanding cannot explain the reduced lamellocyte numbers.

Whereas the authors also present targeted-knock down/inhibition of Pkc53E suggesting that this enzyme is required in blood cells to control crystal cell fate (Fig. 6), it is somehow misleading to use lz-GAL4 as a driver in the lymph gland and hml-GAL4 in circulating hemocytes as these two drivers do not target the same blood cell populations/steps in the crystal cell development process.

We fully agree with the reviewer that the two driver lines target different blood cell populations/ steps in hematopoiesis. The *hml*-Gal4 driver is regarded pan-hemocyte, common to both plasmatocytes and pre-crystal cells (e.g. Tattikota et al., 2020). It has been reported to drive specifically within differentiated hemocytes prior to or at the stage of crystal cells commitment (Mukherjee et al., 2011). Hence, *hml*-Gal4 appeared suitable to hit sessile and circulating hemocytes prior to final differentiation into crystal cells or lamellocytes, respectively.

In the lymph gland, however, *hml* is expressed within the cortical zone, where it appears specific to the plasmatocytes lineage, and not present in the crystal cell precursors (Blanco-Obregon et al., 2020). In contrast, *lz*-Gal4 is specific to the differentiating crystal cells in both lineages, i.e. in circulating and sessile hemocytes and in the lymph gland. Hence, we choose *lz*-Gal4 instead of *hml*-Gal4 at the risk of driving markedly later in the course of crystal cell differentiation. We included the reasoning in the text. Overall, we feel that this choice does not limit our conclusions.

In addition, the authors do not present evidence that Pkc55E function (and Su(H) phosphorylation) is required specifically in blood cells to promote lamellocyte production in response to infestation.

We have tried to address this interesting question by several means. Firstly, we show that Pkc53E is indeed expressed in the various cell types of larval hemocytes, shown in a new Figure 8 and Figure 8_S1 supplement. I.e., there is the potential of Pkc53E to promote lamellocyte formation. Moreover, RNAi-mediated downregulation of Pkc53E within hemocytes affected crystal cell formation similar to the Pkc53ED28 mutant, in agreement with a specific requirement within blood cells (Figure 6). Finally, we show a major drop in Notch target gene transcription (NRE-GFP) in response to wasp infestation within isolated hemocytes from Su(H)gwt in contrast to Su(H)S269A larvae (see new Figure 1 G). These data show that Su(H)-mediated Notch activity must be downregulated in hemocytes prior to lamellocyte formation in agreement with our hypothesis.

Finally, the conclusion that Pkc53E is (directly) responsible for Su(H) phosophorylation needs to be strengthened. Most importantly, the authors do not demonstrate that Pkc53E is required for Su(H) phosphorylation in vivo (i.e. that Su(H) is not phosphorylated in the absence of Pkc53E following infestation).

We would very much like to show respective results. Unfortunately, the low affinity of our pS269 antibody does not allow any in situ or in vivo experiments. We very much hope to obtain a more specific phosphoS269-Su(H) antibody allowing us further in situ studies, and show, for example co-localization with Pkc53E.

In addition, the in vitro kinase assays with bacterially purified Pkc53E (in the presence of PMA or using an activated variant of Pkc53E) only reveal a weak activity on a Su(H) peptide encompassing S269 (Fig. 4).

The reviewer correctly notes the poor activity of our purified Pkc53EEDDD kinase. This low activity also holds true for the standard peptide (PS), which in fact is even less well accepted than the Swt substrate. Indeed, the commercially available PKCα is a magnitude more active. Whether this reflects the poor quality of our isolated protein compared to the commercial PKCα, or whether it reflects a true biochemical property of Pkc53E remains to be shown in the future. We noted this observation in the manuscript.

Moreover, while the authors show a coIP between an overexpressed Pkc53E and endogenous Su(H) (Fig. 7) (in the absence of infestation), it has recently been reported that Pkc53E is a cytoplasmic protein in the eye (Shieh et al. 2023), calling for a direct assessment of Pkc53E expression and localization in larval blood cells under normal conditions and upon infestation.

Indeed, it is interesting that a Pkc53E-GFP fusion protein is cytoplasmic in the eye. The construct reported by Shieh et al. however, i.e. the B-isoform, is preferentially expressed in photoreceptors, where it regulates the de-polymerization of the actin cytoskeleton.

Due to the eye-specific expression, we unfortunately cannot use the Pkc53E-B-GFP construct to test for Pkc53E’s distribution in other tissues.

As this construct is of little use for studying hematopoiesis, we have instead used Pck53E-GFP (BL59413) derived from a protein trap: again, GFP is primarily seen in the cytoplasm of hemocytes, including lamellocytes of infected larvae. However, in a small number of hemocytes, GFP appears to be also nuclear (Fig. 8A), leaving the possibility that activated Pkc53E may localize to the nucleus, eventually phosphorylating Su(H) and downregulating Notch activity. As Su(H) enters the nucleus piggy-back with NICD, however, phosphorylation may as well occur at the membrane or within the cytoplasm. We note, however, that these hypotheses require a much more detailed analysis.

Furthermore, the effect of the PKCa agonist PMA on Su(H)-induced reporter gene expression in cell culture and crystal cell number in vivo is somehow consistent with the authors hypothesis, but some controls are missing (notably western blots to show that PMA/Staurosporine treatment does not affect Su(H)-VP16 level) and it is unclear why STAU treatment alone promotes Su(H)-VP16 activity (in their previous reports, the authors found no difference between Su(H)S269A-VP16 and Su(H)-VP16) or why PMA treatment still has a strong impact on crystal cell number in Su(H)S269A larvae.

We have added a Western blot showing that the treatment does not affect Su(H)-VP16 expression levels (Figure 5_supplement 1). As STAU is a general kinase inhibitor, it may obviate any inhibitory phosphorylation of Su(H)-VP16 in the HeLa cells, e.g. that by Akt1, CAMK2D or S6K which pilot T271, phosphorylation of which is expected to affect the DNA-binding of Su(H) as well (Figure 3_supplement 2). Moreover, in the previous report, we used different constructs with regard to the promoter, and we used RBPJ instead of Su(H), which may explain some of the discrepancies. As PMA is not specific to just Pkc53E, the altered crystal cell numbers may result from the influence on other kinases involved in blood cell homeostasis, as predicted by our genetic screen (Figure 3_supplement 1).

**Reviewer #1 (Recommendations For The Authors):**
(1) The authors should provide a more elaborate examination of larval blood cell types and blood cell counts under normal conditions and following infestation in the different zygotic mutants as well as upon Pkc53 knock-down. A thorough examination of PSC integrity should be performed and the maintenance of core blood cell progenitors examined. The authors should also clarify when after infestation the LG and larval bleeds are analyzed.- a more elaborate examination of larval blood cell types:- examination of larval blood cell counts under normal conditions: hemocyte # in gwt, SA, SD, & Pkc- examination of larval blood cell counts after infestation: hemocyte # in gwt, SA, SD, & Pkc- thorough examination of PSC integrity: in gwt, SA, SD, & Pkc- thorough examination of blood cell progenitors: in gwt, SA, SD, & Pkc- clarify timing

Hemocyte numbers of the various genotypes and conditions were recorded and are presented in Figure 1_S1 and Figure 9_S1. Timing was elaborated in the text and the Methods section.

(2) The authors should clarify why they use lz-GAL4 or hml-GAL4 and what we can infer from using these different drivers.

See above. The reasoning was included in the text.

(3) The percentage of hatching of Su(H)S269A and Su(H)gwt flies in the absence of infestation should also be scored; a small decrease in Su(H)S269A viability might explain the observed differences in survival to wasp infestation. Absolute blood cell numbers (in the absence of infestation) have also been correlated with survival to infection and should be checked.

Percentage of the emerging flies and hemocyte numbers in the absence of infestation were recorded and included in Figure 2, Figure 1_S1, Figure 9_S1.

(4) Whereas the impact of Su(H)S269A or Pkc53E mutation on lamellocytes production is clear, there is still a substantial reduction in crystal cell production following infestation. So I wouldn't conclude that the Su(H) larvae are "unable" to detect this immune challenge or respond to it (line 116).

Thank you for the hint, we corrected the text.

(5) The expression and localization of Pkc53E in larval blood cells should be investigated, for instance using the Pkc53E-GFP line recently published by Shieh et al. (or at least at the RNA level).

Firstly, we confirmed expression of Pkc53E in hemocytes by RT-PCR (Figure 8_S1 supplement). Secondly, expression of Pkc53E-GFP was monitored in hemocytes (Figure 8). To this end, we used the protein trap (BL59413), since the one published by Shieh et al., 2023 is restricted to photoreceptors.

(6) It would be interesting to test the anti-pS269 antibody in immunostaining (using Su(H)S269A as negative control).

Unfortunately, the pS269 antiserum does not work in situ at all.

(7) The authors must perform a western blot with anti-pS269 in Pkc53e mutant to show that Su(H) is not phosphorylated anymore after wasp infestation.

The blot gives a negative result.

(8) It is surprising that no signal is seen in the absence of infestation with anti-pS269: the fact that Su(H)S269A have more crystal cells suggest that there is a constitutive level of phosphorylation of Su(H).

We fully agree: In the ideal world, we would expect a low level of S269 phosphorylation in the wild type as well. However, given the lousy specificity of our antibody, we were happy to see phospho-Su(H) in infected larvae. We are currently working hard to get a better antibody.

(9) The authors should check Su(H)-VP16 levels and phosphorylation status after PMA and/or staurosporine treatment. Some clarifications are also needed to explain the impact of PMA in Su(H)S269 larvae (this clearly suggests that PKC has other substrates implicated in crystal cell development).

Su(H)-VP16 expression levels were monitored by Western blot and were not altered conspicuously (Figure 5_1 supplement). Presumably, Pkc53E is not the only kinase involved in Su(H) phosphorylation or the transduction of stress signals. Moreover, PMA may have a more general effect on larval development and hematopoiesis affecting both genotypes. We included this reasoning in the text.

(10) Concerning the redaction, the authors forgot to mention and discuss the work of Cattenoz et al. (EMBO J 2020). The presentation of the screen for kinase candidates could be streamlined and better illustrated (notably supplement table 4, which would be easier to grasp as a figure/graph). The discussion could be shortened (notably the part on T cells), and I don't really understand lines 374-376 (why is it consistent?).

We are sorry for omitting Cattenoz et al. 2020, which we have now included. We fully agree that this paper is of utmost importance to our work. We streamlined the screen and included a new figure in addition to table 4 summarizing the results graphically (Figure 3_S1 supplement). We cut on the T cell part and omitted the strange lines.

**Reviewer #2 (Public Review):**
Summary:The current draft by Deischel et.al., entitled "Inhibition of Notch activity by phosphorylation of CSL in response to parasitization in *Drosophila*" decribes the role of Pkc53E in the phosphorylation of Su(H) to downregulate its transcriptional activity to mount a successful immune response upon parasitic wasp-infection. Overall, I find the study interesting and relevant especially the identification of Pkc53E in phosphorylation of Su(H) is very nice. However, I have a number of concerns with the manuscript which are central to the idea that link the phosphorylation of Su(H) via Pkc53E to implying its modulation of Notch activity. I enlist them one by one subsequently.Strengths:I find the study interesting and relevant especially because of the following:(1) The identification of Pkc53E in phosphorylation of Su(H) is very interesting.(2) The role of this interaction in modulating Notch signaling and thereafter its requirement in mounting a strong immune response to wasp infection is also another strong highlight of this study.Weaknesses:(1) Epistatic interaction with Notch is needed: In the entire draft, the authors claim Pkc53E role in the phosphorylation of Su(H) is down-stream of notch activity. Given the paper title also invokes Notch, I would suggest authors show this in a direct epistatic interaction using a Notch condition. If loss of Notch function makes many more lamellocytes and GOF makes less, then would modulating Pkc53E (and SuH) in this manifest any change? In homeostasis as well, given gain of Notch function leads to increased crystal cells the same genetic combinations in homeostasis will be nice to see.While I understand that Su(H) functions downstream of Notch, but it is now increasingly evident that Su(H) also functions independent of Notch. An epistatic relationship between Notch and Pkc will clarify if this phosphorylation event of Su(H) via Pkc is part of the canonical interaction being proposed in the manuscript and not a non-canoncial/Notch pathway independent role of Su(H).This is important, as I worry that in the current state, while the data are all discussed inlight of Notch activity, any direct data to show this affirmatively is missing. In our hands we do find Notch independent Su(H) function in immune cells, hence this is a suggestion that stems from our own personal experience.

The role of Notch in *Drosophila* hematopoiesis, notably during crystal cell development in both hematopoietic compartments is well established; likewise the role of Su(H) as integral signal transducer in this context (e.g. Duvic et al., 2002). Not only promotes Notch activity crystal cell fate by upregulating target genes, at the same time it prevents adopting the alternative plasmatocyte fate (e.g. Terriente-Felix et al., 2013). We could confirm the downregulation of Notch target gene expression in response to wasp infestation by qRT-PCR, which was discovered earlier by Small et al. (2014). This is clearly in favor of a repression of Notch activity rather than a relief of inhibition by Su(H). A ligand-independent activation of Notch signaling has been uncovered in the context of crystal cell maintenance in the lymph gland involving Sima/Hif-α, including Su(H) as transcriptional mediator (Mukherjee et al., 2011). However, we are unaware of a respective Su(H) activity independent of Notch.

Certainly, Su(H) acts independently of Notch in terms of gene repression. Here, Su(H) forms a repressor complex together with H and co-repressors Groucho and CtBP to silence Notch target genes. Accordingly, loss of Su(H) or H may induce the upregulation of respective gene expression independent of Notch activity. This has been demonstrated, for example, during wing and heart development (Klein et al., 2000; Kölzer, Klein, 2006; Panta et al., 2020). Moreover, during axis formation of the early embryo, global repression is brought about by Su(H) and relieved by activated Notch (Koromila, Stathopolous, 2019). In all these instances, Su(H) is thought to act as a molecular switch, and the activation of Notch causes a strong expression of the respective genes. Likewise, the loss of DNA-binding resulting from the phosphorylation of Su(H) allows the upregulation of repressed Notch target genes in wing imaginal discs, e.g. *dpn*, as we have demonstrated before with overexpression and clonal analyses (Nagel et al. 2017; Frankenreiter et al., 2021). However, H does not contribute to crystal cell homeostasis, i.e. de-repression of Notch target genes does not appear to be a major driver in this context, asking for additional mechanisms to downregulate Notch activity. Our work provides evidence that these inhibitory mechanisms involves the phosphorylation of Su(H) by Pkc53E. Formally, we cannot exclude alternative mechanisms. Hence, we have tried to avoid the direct link between Su(H) phosphorylation and the inhibition of Notch activity throughout the text, including the title. Moreover, we have discussed the possible consequences of Su(H) lack of DNA binding, interfering either with the activation of Notch target genes or abrogating their repression.

In addition, we have performed new experiments addressing the epistasis between Notch and Su(H) during crystal cell formation (Figure 1_supplement 1). To this end, we knocked down Notch activity in hemocytes by RNAi (hml::N-RNAi) in the Su(H)gwt and Su(H)S269A background, respectively. Indeed, Notch downregulation strongly impairs crystal cell development independent of the genetic background as expected if Notch were epistatic to Su(H). We attribute the slightly elevated crystal cell numbers observed in the Su(H)S269A background to the increase in the embryonic precursors (see Fig. 4; Frankenreiter et al. 2021). Of note, the Notch gain of function allele Ncos479 also displayed a likewise increase in embryonic crystal cell precursors as well as in crystal cells within the lymph gland (Frankenreiter et al. 2021).

(2) Temporal regulation of Notch activity in response to wasp-infection and its overlapping dynamics of Su(H) phosphorylation via Pkc is needed:

First, I suggest the authors to show how Notch activity post infection in a time course dependent manner is altered. A RT-PCR profile of Notch target genes in hemocytes from infected animals at 6, 12, 24, 48 HPI, to gauge an understanding of dynamics in Notch activity will set the tone for when and how it is being modulated. In parallel, this response in phospho mutant of Su(H) will be good to see and will support the requirement for phosphorylation of Su(H) to manifest a strong immune response.

Indeed, it would be extremely nice to follow the entire processes in every detail, ideally at the cellular level. The challenge, however, is quantities. The mRNA isolated from hemocytes could be barely quantified, although the subsequent ct-values were ok. We quantified NRE-GFP expression, introduced into Su(H)gwt and Su(H)S269A, as well as atilla expression. We were able to generate data for two time slots, 0-6 h and 24-30 h post infection. The data are provided in the extended Figure 1G, and show a strong drop of NRE-GFP in the infected Su(H)gwt control compared to the uninfected animals, whereas expression in Su(H)S269A plateaus at around 60%-70% of the infected Su(H)gwt control. Atilla expression jumps up in the control, but stays low in Su(H)S269A hemocytes.

Second, is the dynamics of phosphorylation in a time course experiment is missing. While the increased phosphorylation of Su(H) in response to wasp-infestation shown in Fig.2B is using whole animal, this implies a global down-regulation of Su(H)/Notch activity. The authors need to show this response specifically in immune cells. The reader is left to the assumption that this is also true in immune cells. Given the authors have a good antibody, characterizing this same in circulating immune cells in response to infection will be needed. A time course of the phosphorylation state at 6, 12, 24, 48 HPI, to guage an understanding of this dynamics is needed.

We really would love to do these experiments. Unfortunately, our pS269 antibody is rather lousy. It does not allow to detect Su(H) protein in tissue or cells, nor does it work on protein extracts in Westerns or for IP. Hence, we have no way so far to demonstrate cell or tissue specificity of Su(H) phosphorylation. So far, we were lucky to detect mCherry-tagged Su(H) proteins pulled down in rather large amounts with the highly specific nano-bodies. We have tried very hard to repeat the experiment with hemolymph and lymph glands only, but we have failed so far. Hence, we have to state that our antibody is neither suitable for in vivo analyses, nor for a detection of phospho-Su(H) at lower levels.

The authors suggest, this mechanism may be a quick way to down-regulate Notch, hence a side by side comparison of the dynamics of Notch down-regulation (such as by doing RT-PCR of Notch target genes following different time point post infection) alongside the levels of pS269 will strengthen the central point being proposed.

We fully agree and hope to address these issues in the future by improving our tools.

Last, in Fig7. the authors show Co-immuno-precipitation of Pkc53EHA with Su(H)gwt-mCh 994 protein from Hml-gal4 hemocytes. I understand this is in homeostasis but since this interaction is proposed to be sensitive to infection, then a Co-IP of the two in immune cells, upon infection should be incorporated to strengthen their point.

We do not fully agree with the reviewer. Although we also think that the interaction between Pkc53E and Su(H) might occur more frequently upon infection, we propose that this is a transient process occurring in several but not all hemocytes at a given time. Moreover, in the described experiment, Pkc53E-HA was expressed in hemocytes via the UAS/Gal4 system. We cannot exclude that this approach causes an overexpression. Hence, we would not expect considerable differences between unchallenged and infested animals.

(3) In Fig 5B, the authors show the change in crystal cell numbers as read out of PMA induced activation of Pkc53E and subsequent inhibition of Su(H) transcriptional activity, I would suggest the authors use more direct measures of this read out. RT-PCR of Su(H) target genes, in circulating immune cells, will strengthen this point. Formation of crystal cells is not just limited to Notch, I am not convinced that this treatment or the conditions have other affect on immune cells, such as any impact on Hif expression may also lead to lowering of CC numbers. Hence, the authors need to strengthen this point by showing that effects are direct to Notch and Su(H) and not non-specific to any other pathway also shown to be important for CC development.

We agree with the Reviewer that the rather general influence of PMA on PKCs might present a systemic stress to the animal. For example, we observed a slight drop of crystal cell numbers also in Su(H)S269A, suggesting other kinases apart from Pkc53E were affected that are involved in crystal cell homeostasis. We have included this notion in the text. To provide more conclusive evidence we also fed Staurosporine to the larvae which reversed the PMA effect. In addition, we assayed the expression of NRE-GFP in hemocytes of infected animals by qRT-PCR, and observed a strong drop in the infected versus uninfected control but less so in Su(H)S269A. The new data are provided in extended Figures 1G and 5B.

(4) In addition to the above mentioned points, the data needs to be strengthened to further support the main conclusions of the manuscript. I would suggest the authors present the infection response with details on the timing of the immune response. Characterization of the immune responses at respective time points (as above or at least 24 and 48 HPI, as norms in the field) will be important. Also, any change in overall cell numbers, other immune cells, plasmatocytes or CC post infection is missing and is needed to present the specificity of the impact. The addition of these will present the data with more rigor in their analysis.

Total hemocyte numbers of the various genotypes, i.e. control, Su(H)S269A, Su(H)S269D, and Pkc53ED28 were included before and after wasp infestation in supplemental Figures 1_S1 and 9_S1.

(5) Finally, what is the view of the authors on what leads to activation of Pkc53E, any upstream input is not presented. It will be good to see if wasp infection leads to increased Pkc53 kinase activity.

The analysis of the full process is an ongoing project. We propose that ROS is produced upon the wasps’ sting, which is to trigger the subsequent cascade of events. These have to end with activation of Pkc53E in the presumptive pre-lamellocyte pool of both lineages, i.e. in plasmatocyte of the hemolymph, presumably in the sessile compartment (Tattikotta et al., 2021) and at the same time in the lymph gland cortex harboring the LM precursors (Blanco-Obregon et al., 2020). One of the known upstream kinases, Pdk1 has a similar impact on crystal cell development as Pkc53E, making its involvement likely. Moreover, we think that other PKCs influence the process as well.

Without a good read out, e.g. a functional pSu(H) antiserum working in situ or a Pkc-activity reporter, it will be quite difficult to follow up this question. However, we already know that Pkc53E is expressed in hemocytes of all types independent of wasp infestation, in agreement with a role during lamellocyte differentiation. We hope to unravel the process in more of it in the future.

Overall, I think the findings in the current state are interesting and fill an important gap, but the authors will need to strengthen the point with more detailed analysis that includes generating new data and also presenting the current data with more rigor in their approach. The data have to showcase the relationship with Notch pathway modulation upon phosphorylation of CSL in a much more comprehensive way, both in homeostasis and in response to infection which is entirely missing in the current draft.
**Reviewer #3 (Public Review):**
Diechsel et al. provide important and valuable insights into how Notch signalling is shut down in response to parasitic wasp infestation in order to suppress crystal cell fate and favour lamellocyte production. The study shows that CSL transcription factor Su(H) is phosphorylated at S269A in response to parasitic wasp infestation and this inhibitory phosphorylation is critical for shutting down Notch. The authors go on to perform a screen for kinases responsible for this phosphorylation and have identified Pkc53E as the specific kinase acting on Su(H) at S269A. Using analysis of mutants, RNAi and biochemistry-based approaches the authors convincingly show how Pkc53E-Su(H) interaction is critical for remodelling hematopoiesis upon wasp challenge. The data presented supports the overall conclusions made by the authors. There are a few points below that need to be addressed by the authors to strengthen the conclusions:(1) The authors should check melanized crystal cells in Su(H)gwt and Su(H)S269A in presence of PMA and Staurosporine?

Thank you for the suggestion. We included the results of PMA + Staurosporine feeding into an extended Fig. 5B; they match those from the HeLa cells. Unfortunately, Staurosporine alone was lethal for the larvae at various concentrations, presumably owing to the overarching inhibition of kinase activity. This global effect also explains the high crystal cell numbers in the control fed with PMA + STAU compared to the untreated animals, as the downregulation of many kinases results in higher crystal cell numbers, a fact uncovered in our genetic screen.

(2) Data for number of dead pupae, flies eclosed, wasps emerged post infestation should be monitored for the following genotypes and should be included:

Pkc53EΔ28_, Su(H)S269A,_ Pkc53EΔ28 Su(H)S269A, Su(H)S269D, Su(H)S269D Pkc53EΔ28

We extended the data with and without infection. The respective data are shown in a new Fig. 9 and an extended Fig. 2, except for the Su(H)S269D allele. Su(H)S269D is larval lethal, i.e. dies too early for wasp development, and hence could not be included in the assay. Overall, Pkc53EΔ28 matched Su(H)S269A_._

(3) The exact molecular trigger for activation of Pkc53E upon wasp infestation is not clear.

Indeed, and we would love to know! Perhaps, the generation of Ca2+ by the wasp’s breach of the larval cuticle results in Pkc53E activation. The generation of ROS could be involved as well. At this point, we can only speculate. We hope to be able in the future to obtain direct experimental evidence for the one or the other hypothesis.

(4) The authors should check if activating ROS alone or induction of Calcium pulses/DUOX activation can mimic this condition and can trigger activation of Pkc53E and thereby cause phosphorylation of Su(H) at S269

The reviewer’s suggestions open up a new field of investigations, and are hence beyond of the scope of this article. However, we want to pursue the research in this direction, albeit we realize that counting crystal cells is too coarse but to give a first impression, and that lamellocytes may form already by breaching the larval cuticle. A major challenge shall be direct measurements of Pkc53E activation. To date, we have no tools for this, but ideally, we would like to have a direct, biochemical read out. Although we have been unsuccessful in the past, we want to develop a strong and specific phospho-S269 antibody that is also working in situ. Alternatively, we think of developing a PS-phosphorylation reporter, to allow reasonably addressing these questions.

(5) Does Pkc53E get activated during sterile inflammation?

We are in the process of addressing this issue, however, feel that his topic is beyond the scope of this paper. Our preliminary experiments, however, support the notion of a phospho-dependent regulation of Su(H) also in this context.

**Reviewer #3 (Recommendations For The Authors):**
The authors provide a graphical representation of major phenotypes that form the basis of their investigation and conclusions but have not supplemented the quantitation with images that represent these phenotypes. The authors need to include the following data to strengthen their conclusions:(1) The authors should include representative images for each of the genotypes/conditions (in presence and absence of wasp infestation) based on which corresponding plots have been made in Figure 1. Please include this for both circulating lamellocytes in the hemolymph and in the lymph glands since this is one of the main figures presenting the key findings.

The data have been included in Figure 1-S2 supplement.

(2) Please include representative images of LG with Hnt staining and corresponding images for melanization for each of the genotypes used in the plots in Figure 6A and B.

The data have been included in Figure 6-S2 supplement.

(3) Representative images for each of the genotypes in Figure 7A & B should be included (circulating crystal cells and lymph gland crystal cell numbers).

Representative images for each of the genotypes for Fig. 7A have been included in Figure 7-S1 and for the old Fig. 7B in Figure 9-S2 supplement, respectively.